# Hydroethanolic extract of *Schinus terebinthifolia* as a promising source of anti-influenza agents: Phytochemical profiling, cheminformatics, molecular docking and dynamics simulations

Napapuch Nopkuesuk[1◉], Anuwatchakij Klamrak[1◉], Jaran Nabnueangsap[2], Jaraspim Narkpuk[3], Shaikh Shahinur Rahman[1,4], Yutthakan Saengkun[1,5], Piyapon Janpan[1,5], Thananya Soonkum[2], Poramet Sitthiwong[6], Nisachon Jangpromma[5,7], Sirinan Kulchat[8], Kiattawee Choowongkomon[9], Rina Patramanon[5,7], Arunrat Chaveerach[10], Samaporn Teeravechyan[3], Jureerut Daduang[11], Sakda Daduang[1,5*]

1 Division of Pharmacognosy and Toxicology, Faculty of Pharmaceutical Sciences, Khon Kaen University, Khon Kaen, Thailand, 2 Salaya Central Instrument Faculty RSPG, Research Management and Development Division, Mahidol University, Nakhon Pathom, Thailand, 3 Virology and Cell Technology Research Team, National Center for Genetic Engineering and Biotechnology (BIOTEC), National Science and Technology Development Agency (NSTDA), Pathumthani, Thailand, 4 Department of Applied Nutrition and Food Technology, Faculty of Biological Sciences, Islamic University, Kushtia, Bangladesh, 5 Protein and Proteomics Research Center for Commercial and Industrial Purposes (ProCCI), Khon Kaen University, Khon Kaen, Thailand, 6 Khaoyai Panorama Farm Co., Ltd., Nakhonratchasima, Thailand, 7 Department of Biochemistry, Faculty of Science, Khon Kaen University, Khon Kaen, Thailand, 8 Department of Chemistry, Faculty of Science, Khon Kaen University, Khon Kaen, Thailand, 9 Department of Biochemistry, Faculty of Sciences, Kasetsart University, Bangkok, Thailand, 10 Department of Biology, Faculty of Science, Khon Kaen University, Khon Kaen, Thailand, 11 Department of Clinical Chemistry, Faculty of Associated Medical Sciences, Khon Kaen University, Khon Kaen, Thailand

◉ These authors contributed equally to this work.
* sakdad@kku.ac.th

## Abstract

Although *Schinus terebinthifolia* (commonly known as Brazilian peppertree) has been documented to possess various biological activities, such as anticancer, antibacterial, and antioxidant properties, its anti-influenza activity has not yet been documented. Here, an aqueous-ethanolic extract (30% v/v ethanol solution), prepared from its aerial parts (leaves and stalks), was established to determine whether it is a rich source of antiviral agents. The hydroethanolic plant extract, with a TPC value of 264.11 mg (GAE)/g DE, exhibits a promising $IC_{50}$ value of 16.33 µg/mL, similar to that of authentic quercetin ($IC_{50} = 12.72$ µg/mL), and approximately 5.34 times higher than that of gallic acid ($IC_{50} = 3.06$ µg/mL) as determined by the DPPH assay. This extract contains 1.71 mg of gallic acid (representative marker) per gram of dried plant material, according to HPLC analysis. Using untargeted metabolomics analysis coupled with a series of cheminformatics tools (MetFrag, SIRIUS, CSI:FingerID, and CANOPUS), we ultimately proved that the *S. terebinthifolia* hydroethanolic extract contains simple phenolics (e.g., methyl gallate, ethyl gallate, and chlorogenic acid), flavonoids

**Data availability statement:** All relevant data are within the paper.

**Funding:** This research was funded by the Program Management Unit for Human Resources and Institutional Development, Research and Innovation (PMU-B) under a post-master's scholarship (PMU-B grant number: B13F660069). Partial funding was also provided by the Fundamental Fund of Khon Kaen University (KKU), supported by the National Science, Research and Innovation Fund (NSRF), Thailand. Additional support was received from the Basic Research Fund of Khon Kaen University, under the NSRF.

**Competing interests:** No authors have competing interests.

(afzelin and myricitrin), dicarboxylic acids, and germacrone. As anticipated, the plant extract exhibited anti-influenza activity with an $IC_{50}$ of 2.21 µg/mL ($CC_{50} > 50$ µg/mL) and did not exert hemolytic activity at the concentration of 2000 µg/mL, underscoring its efficacy as a safe antiviral solution. *In silico* molecular docking and dynamic simulations suggest that neuraminidase and the cap-binding domain of influenza RNA polymerase (PB2) are preferentially targeted for inhibition by the detected metabolites. Owing to the diverse therapeutic effects of secondary metabolites, the anti-H5N1 activity of the newly developed plant extract is currently under investigation.

## 1. Introduction

Humans are vulnerable to a variety of viral illnesses, one of which is H1N1, which poses a significant threat to public health [1]. The high morbidity and mortality rates associated with this disease make it a pressing issue that requires attention. Viral diseases are not limited to a specific geographic region and can cause severe health problems in both children and the elderly. Although vaccines are available, influenza still causes an estimated 250,000–500,000 deaths annually in industrialized countries, with 6.1 deaths per 100,000 population reported in Thailand, partly due to the virus's ability to develop drug resistance through mutations [2,3]. Hence, there is a need for new medicines, and plants offer a promising source due to their rich secondary metabolites with antioxidant and broad therapeutic potential against both infectious and non-infectious diseases.

Several studies have shown that plant extracts, including aqueous and ethanolic extracts, can be a promising source of antiviral agents [4]. Various stages of the viral life cycle, including attachment, entry, replication, spread, and particle maturation, can be disrupted by plant extracts due to polyphenols, hydroxycinnamic acids, and phenylpropanoid pathway products [5,6]. For example, *Toona sinensis* is a plant that contains gallic acid and catechin, which has been found to inhibit various stages of influenza virus replication [7]. *In silico* studies also suggest that these compounds interact with specific sites on the virus's neuraminidase (NA), hemagglutinin (HA), and RNA polymerase. Similarly, phenolic compounds found in *Aronia melanocarpa*, including isoquercetin, kaempferol, ferulic acid, caffeic acid, ellagic acid, and myricetin, have been shown to have antiviral properties against influenza viruses [8]. Furthermore, these compounds have been found to improve the survival rates of mice, infected with the rPR8-GFP virus due to their antioxidant and anti-inflammatory properties [9,10]. Our recent study shows that the aqueous extract of *C. mimosoides*, which primarily consists of gallic acid and related substances such as quinic acid, shikimic acid, and protocatechuic acid, can inhibit influenza A (H1N1) virus infection in MDCK cells with an $IC_{50}$ of 5.14 µg/mL [11]. This brought us to seeking, therefore, to identify and develop extracts from other plant species that contain a large number of simple phenolics and flavonoids as potential antiviral agents, while minimizing organic solvent consumption through green chemistry approaches, which can be scaled up for industrial applications.

*Schinus terebinthifolia*, often referred to as the "Brazilian pepper tree," is a plant belonging to the Anacardiaceae family that is utilized as a traditional medicine to treat various ailments, such as rheumatism, hypertension, ulcers, gastric distress, menstrual disorders, gonorrhea, bronchitis, gingivitis, conjunctivitis, dysentery, wounds, urinary tract infections, and eye infections [12]. Phytochemical analyses have demonstrated that it contains a diverse range of natural products, including phenolic acids, flavonoids, and terpenoids, which have been relevant to the observed antibacterial, antifungal, insecticidal, antiparasitic, analgesic, cytotoxic, antitumor, antioxidant, antihypertensive, anti-inflammatory, antimycobacterial, anti-Parkinson, anti-allergic, antiviral, wound healing, chemoprotective, anthelmintic, and hepatoprotective properties [13,14]. Although *S. terebinthifolia* is native to South America, including Chile, Argentina, Brazil, and Paraguay [15], it has also been widely distributed in northeastern Thailand, where its young leaves are commonly consumed with Northeastern-style Esan dishes, such as spicy minced pork and papaya salads. The documented pharmacological properties and local availability suggest that *S. terebinthifolia's* leaves have potential as a new antiviral agent, despite no studies having been conducted on this topic to date.

This study investigated the antiviral potential of a hydroethanolic extract of *S. terebinthifolia*, inspired by prior findings that a 30% (v/v) aqueous-ethanol extract of *C. mimosoides* was more effective than aqueous extracts [11,16]. The extract was analyzed for total phenolic content (TPC), antioxidant activity, HPLC, and untargeted metabolomics (UPLC-ESI(±)-QTOF-MS/MS) to explore its phytochemical profile, which was annotated using in-silico tools (MetFrag, Sirus, CSI:FingerID, CANOPUS), achieving high confidence (level 2) according to metabolomics guidelines [17]. SwissADME assessed drug-likeness and physicochemical properties, confirming the therapeutic potential of the identified molecules [18]. The extract was tested for anti-influenza A/PR/8/34 virus and hemolytic activity, and molecular docking and MD simulation were conducted to explore its antiviral mechanisms.

## 2. Materials and methods

### 2.1. Preparation of *S. terebinthifolia* aqueous ethanol extract

Fresh materials of *S. terebinthifolia*, including leaves and stalks, were harvested from an open field in Nam Phong subdistrict, Khon Kaen province, Thailand (16°37′59.9″ N, 102°46′14.3″ E). The plant is locally available; therefore, no special permission was required for its collection. The collected plant materials were rinsed thoroughly with tap water multiple times and then soaked twice in deionized water for 60 min each to ensure all residues were removed. The plant materials were dried at 50°C for 2–3 days in an oven incubator (Model KT(E6), BINDER, Neckarsulm, Germany). Once completely dried, they were ground into a fine powder using a blender. For the hydro-ethanolic extraction, 20 g of the powdered plant material was placed in a 500 mL Erlenmeyer flask and extracted with 200 mL of 30% (v/v) hydroethanolic solution. The extraction was performed by gently shaking at 100 rpm at 25°C for 48 hours in an incubator shaker (Model NB-250VL, N-BIOTEK, Gyeonggi-do, South Korea). The supernatant was separated from the plant extract at 25°C and 8,000 rpm for 20 min using ST1R Plus-MD Refrigerated Centrifuge (Thermo Scientific, Massachusetts, USA). The clear supernatant was transferred to a round-bottom flask, and approximately 60 mL of ethanol was removed using a Rotavapor R-3 Laboratory Evaporator (BUCHI, Finland). The 140 mL water-based solution was freeze-dried at −110°C for 48 hours using a CoolSafe freeze dryer (Labogene, Allerød, Denmark) and stored at −80°C until analysis.

### 2.2. Total Phenolic Content (TPC)

Total phenolic content was determined as described by Folin and Ciocalteu (1927) with appropriate modifications [19,20], in which the reaction was done in a 96-well plate. An aliquot (20 μL) of *S. terebinthifolia* extract and authentic gallic acid (Product No. G7384, Sigma-Aldrich, Missouri, USA). Next, 100 μL of 0.2 M Folin–Ciocalteu reagent and 80 μL of 7% (w/v) sodium carbonate were added. The 30% v/v ethanol solution was used as a blank. The mixtures were incubated at 25°C for 30 min before being measured using a microplate reader at 760 nm (Ensight Multimode Plate Reader, PerkinElmer,

Massachusetts, USA). Phenolic content in the aqueous-ethanolic extract was quantified as gallic acid equivalents (GAE) or quercetin equivalents (QAE) per milligram of dry plant material, using standard calibration curves.

## 2.3. Antioxidant activity assay

The 2,2-diphenyl-1-picrylhydrazyl (DPPH) assay was performed to evaluate the radical scavenging activity of the newly established extract, as this property has also been reported to be associated with antiviral activities [21,22]. The protocol was implemented as described by Xiao et al. [23]. Briefly, 100 µL of the diluted plant extract, with concentrations ranging from 7.8 to 1000 µg/mL, was mixed individually with 100 µL of 0.2 mM DPPH reagent and incubated in the dark for 30 min. Authentic gallic acid and quercetin were used as reference standards, with the aqueous ethanolic solution (30% v/v ethanol) serving as the blank sample. The decrease in absorbance was measured at 517 nm using a microplate reader (Ensight Multimode Plate Reader). The obtained results were expressed as $IC_{50}$ values and the percentage of DPPH radical scavenging activity. The inhibition ratio (%) was calculated using the formula: inhibition ratio (%) = [($A_{control}$ - $A_{sample}$)/ $A_{control}$] × 100, where $A_{control}$ represents the absorbance of the blank sample and $A_{sample}$ represents the absorbance of the hydroethanolic plant extract.

## 2.4. HPLC determination of gallic acid

Determination of gallic acid, as a representative marker in the *S. terebinthifolia* hydroethanolic extract, was carried out because this compound has been reported to inhibit various stages of the influenza life cycle, including spreading and replication [24]. HPLC system (Agilent Technologies 1260 Infinity) connected with a UV detector was used. Metabolite separation was achieved using a C-18 column (100 mm × 4.6 mm, 5 µm particle size, Phenomenex, California, USA). Two mobile phases consisted of solvent A (0.5% phosphoric acid) and solvent B (methanol), where separation of the target metabolite was achieved using a linear gradient of solvent B, starting with 5% B for 2 min, increasing to 95% B over 15 min, holding at 95% B for 3 min, and returning to 5% B for the final 10 min, with a total run time of 30 min. The flow rate was maintained at 0.8 mL/min. Both the sample and column were kept at a temperature of 25°C. Detection of gallic acid was performed at 270 nm (maximum absorbance), and its quantification was based on a five-point standard curve of gallic acid (62.5–1000 µg/mL) with the equation $y = 68.215x + 1534.2$ ($R^2 = 0.9976$). The yield was expressed as mg/g of dried plant material.

## 2.5. UPLC-ESI(±)QTOF-MS/MS

Besides identifying gallic acid, phytochemical profiling was established using both negative and positive ion mode analyses to fully detail the other bioactive constituents presumably responsible for the antioxidant and anti-influenza activities present in the *S. terebinthifolia* aqueous ethanolic extract. An ultrahigh performance liquid chromatograph (UHPLC) (UltiMate 3000 RSLCnano UHPLC System, Thermo Scientific, Massachusetts, USA), which is equipped with Acclaim RSLC120 C18 (100 × 2.1 mm, 2.2 µm 120Å, Thermo Scientific, USA) column were used in the separation processes. The 0.1% formic acid (solvent A) and acetonitrile (solvent B) were the mobile phases with a flow rate of 0.4 mL/min. A linear gradient of the solvent B as following conditions: 2% for 0–2 min, 0–90% for 1–12 min and hold for 2 min, and back to 2% for 4 min for a total run time of 18 min was implemented. The 1 µL of the aqueous ethanolic extract (10 ppm) was separated through the RP column, where the temperature was maintained at 35°C. Metabolite identification of a mass spectrometer (TripleTOF6600 +, AB SCIEX, Massachusetts, USA) connected with the HPLC instrument. ESI source conditions were as follows: the ion source gas 1 50, ion source gas 2 60, curtain gas 30, a temperature of 150 °C, with the ion spray voltage floating of −4500 V in negative mode and 5500 V in positive mode. A TOF MS scan range of 100–800 amu. The product ion scan range of 50–800 Da. The scan accumulation time was 0.2 s, while the parent ion scan accumulation time was 0.25. The mass spectra were generated using a delustering potential (DP) of 80 V with the collision energy of 40 ± 10 eV. Structural matching was based on spectral comparison with authentic MS/MS spectra available

in the Natural Products HR-MS/MS library and NIST 2017, which contains approximately 13,800 substances. The raw $MS^2$ data of various suspected metabolites were further annotated using mass spectral annotation tools to reach high-confidence metabolite identification, adhering to the guidelines of the Metabolomics Standards Initiative (MSI), as described by Sumner et al. [25].

## 2.6. Structural annotation using MetFrag

The MetFrag web service (https://msbi.ipb-halle.de/MetFrag/) was utilized to annotate the acquired raw mass data, providing detailed information on the elemental formula, mass, and potential structure of the queried subjects. According to the official user guidelines (https://ipb-halle.github.io/MetFrag/projects/metfragweb/), the MetFrag workflow consists of two main steps: (i) retrieving candidate structures from relevant databases and (ii) configuring and processing the fragmentation data. Initially, the molecular formula and corresponding m/z value of the target compound were specified. Suspected biological databases (e.g., KEGG and NORMAN) were recommended over larger databases, such as PubChem, to ensure the accuracy of the annotation. Second, the raw MS/MS data were compared to the in-silico generated spectra of the retrieved structural candidates. Candidate structures with an F score close to or equal to 1.0 were identified as potential matches for the query metabolites.

## 2.7. Metabolite annotation using SIRIUS (v.5.6.8)

Since Sirius has been developed to work with CSI:FingerID and CANOPUS to provide more structural information (fingerprints) and classifications of the query metabolites, this software was chosen to fully detail the bioactive agents detected in the *S. terebinthifolia* aqueous ethanolic extract, following the procedure described by Ludwig et al. [26]. To begin with, the query raw mass data was imported into the software, followed by setting various parameters such as collision-induced dissociation (CID) energy, specifying the precursor ion (e.g., *m/z* 197.0522), and selecting the adduct type. In this step, the anticipated formula (e.g., $C_9H_{10}O_5$) can be provided to enhance the accuracy of metabolite annotation. After computing, SIRIUS, CSI-FingerID, and CANOPUS were all selected to gather more detailed information on natural products. Similar to MetFrag, narrowing it down to biological databases such as KEGG, NORMAN, PlantCyc, and Natural Products improves the accuracy of metabolite annotation, particularly when the query metabolites originate from biological samples.

## 2.8. Anti-influenza virus screening assay

Here, we chose the influenza virus strain A/Puerto Rico/8/34 (H1N1) as a surrogate viral strain to ensure that the *S. terebinthifolia* extract can inhibit its life cycle in MDCK cells. The assay was carried out in a 96-well plate. First, MDCK cells were seeded at 1.5x105 cells/well in serum-free OptiMEM (Invitrogen, Massachusetts, USA) in the presence of 2 µg/mL TPCK-treated trypsin and incubated at 5% $CO_2$ at 37°C for 24 hours. The plant extract, at concentrations ranging from 1.56 to 200 µg/mL, was equally mixed (1:1) with viral particles (final concentration of 100 pfu/well), incubated at 25°C for 1 hour, and then treated to MDCK cells for another 24 hours. After fixing with the ice-cold acetone for 15 min, the treated MDCK cells were blocked with PBS containing 0.5% Tween 20 (PBST) and 2% BSA and incubated at 25°C for 30 min. For the detection assay, primary mouse anti-influenza A NP-UNLB antibody (BioLegend, California, USA), diluted 1:2000 in PBST with 1% BSA, was added to the fixed cells and incubated for 60 min. After washing the PBS, HRP-conjugated goat anti-mouse antibody (BioLegend, California, USA) was added (1:10,000 dilution), and the reactions were incubated for 60 min. For colorimetric detection, 3,3′,5,5′-Tetramethylbenzidine (TMB) functioning as a substrate was added, where the reaction was proceeded at room temperature for 8 min. The signal was measured 450 nm. Prior to this experiment, an MTT assay was performed to assess cytotoxicity, optimize extract concentration (100 to 0.05 µg/mL) [27], evaluate safety, and compare baseline cytotoxicity, ensuring the distinction between the extract's effects on cell viability and its specific antiviral activity.

## 2.9. Hemolytic activity assay

The hemolytic potential of the *S. terebinthifolia* aqueous ethanolic extract was then investigated towards human red blood cells (hRBCs) to assess its safety profile as recently established by our group [28]. The research protocol was approved by the Ethics Committee of the Faculty of Biological Sciences, Islamic University, Kushtia, Bangladesh (Approval No. ERC/FBS/I.U./2022/09). Before blood collection, written informed consent was obtained from the participant, and all procedures were conducted following the Declaration of Helsinki's guidelines and regulations. In brief, the newly collected hRBCs were gently resuspended in PBS (pH 7.4) to a final concentration of 4% (v/v). The suspension was then pipetted into a new tube, mixed with the plant extract at concentrations ranging from 15.63 to 2,000 μg/mL, and incubated at 37°C for 60 min. Triton-X 100 (final concentration of 0.1%) served as the positive control. The mixtures were then centrifuged at 10,000×g for 5 min. The clear supernatant was transferred to a 96-well plate, and the signal intensity was measured at 415 nm. Hemolytic activity was determined using the equation: % hemolysis = (S/P) × 100. In this equation, S refers to the absorbance of the sample (e.g., extract and PBS), and P refers to the absorbance of the positive control, 0.1% Triton X-100. A hemolytic activity under 15% reveals a favorable safety profile for the plant extract, suggesting its potential for human use, as mentioned in the literature [29].

## 2.10. Molecular docking

*In silico* docking studies were conducted to elucidate the mechanisms underlying the antiviral activity of the *S. terebinthifolia* hydroethanolic extract. Hemagglutinin (HA; PDB: 1RU7), neuraminidase (NA; PDB: 6HP0), and the cap-binding protein of influenza RNA polymerase (PB2; PDB: 1RU7) were chosen due to their significant roles in the cellular entry, spread, and replication of influenza viruses, respectively [16]. All protein structures were retrieved from the Protein Data Bank (https://www.rcsb.org/), and molecular docking was performed using GOLD (Genetic Optimization of Ligand Docking) software, version 5.2.2. After performing a self-docking experiment, the crystal structure of NA co-crystallized with the GJT ligand achieved an RMSD value of 1.2578, indicating its validity for docking purposes. The HA structure, with a docking pose generated by sialic acid using CB-DOCK (https://cadd.labshare.cn/cb-dock2/php/index.php), exhibited a promising RMSD value of 0.7098. The 4NCE structure, which is bound to 7-methyl-GTP, was also found to exhibit a valid RMSD value of 1.7851. The Swiss PDB Viewer (v.4.1.0) was used to repair the missing residues in the crystal structures of NA and 4NCE to generate more reliable results. All 3D ligand structures in SDF format were obtained from the PubChem database (https://pubchem.ncbi.nlm.nih.gov/).

## 2.11. Molecular dynamic simulation

The chosen protein–ligand complexes of two target proteins and their ligands, obtained from the GOLD program, were simulated using the GROMACS simulation package through the SiBioLead online molecular dynamics simulation web server (https://sibiolead.com/MDSIM) to investigate the dynamic behavior, stability, and interactions of the complexes over the simulation time. The simulation analyses were conducted in four sequential stages. In the preprocessing stage, topologies were defined using the AMBER99SB force field in a cubic box with the SPC (Simple Point Charge) water model, and the system was neutralized with 0.15 M NaCl. In the energy minimization stage, the steepest descent algorithm was used as the EM integrator with 5,000 steps. During the equilibration stage, the system was simulated using the NVT/NPT method at 1.0 bar pressure and 37°C for 100 ps. The molecular dynamics simulation was then run for 100 ns using the leap-frog integrator, and the trajectories were analyzed using the standard GROMACS analysis tools.

## 3. Results and discussion

### 3.1. TPC and Radical Scavenging Activity of plant extract

The available evidence suggests a strong link between the phenolic compounds found in specific plant extracts and their antiviral properties, as these compounds may interact with the active sites of viral proteins, such as the NA, HA, and

PB2-subunit, by forming strong hydrogen bonds and hydrophobic interactions [22]. Hence, it is essential to evaluate the radical scavenging activity and total phenolic content (TPC) prior to conducting further experiments. The TPC value of the hydroethanolic plant extract was 264.11 mg (GAE)/g DE. According to the DPPH assay, its $IC_{50}$ value of 16.33 µg/mL is comparable to that of authentic quercetin ($IC_{50}$ = 12.72 µg/mL) and approximately 4.91 times higher than standard gallic acid ($IC_{50}$ = 3.06 µg/mL), which implies potential antiviral activity (Table 1). These results might imply the presence of both simple phenolics and flavonoids However, further investigation of its phytochemical contents is necessary.

### 3.2. HPLC analysis of gallic acid in *S. terebinthifolia* hydroethanolic extract

Gallic acid is proposed to contribute to anti-influenza activities by inhibiting multiple stages, including early stages and replication [24,30]. Therefore, it was chosen as a representative marker for the quality control of the *S. terebinthifolia* aqueous ethanolic extract. A peak (Rt = 10.054 min) matching that of authentic gallic acid (Rt = 10.096 min) was identified in the 30% hydroethanolic extract (Fig 1). Using a five-point calibration curve ($R^2$ = 1), the yield of gallic acid was determined to be 1.71 mg/g DW (dried weight) of plant material, suggesting it potentially be used as a new anti-influenza agent. Since the gallic acid peak (Rt = 10.054 min) accounts for 12.24% of the relative peak area, the remaining substances, approximately 87.76%, require further exploration to identify other polyphenols and other compounds in the extract that may synergistically contribute to its antioxidant and anti-influenza properties.

### 3.3. Untargeted metabolomics and computationally assisted identification of bioactive metabolites detected from the *S. terebinthifolia* hydroethanolic extract

The use of untargeted metabolomics analyses, both in negative and positive ion modes, led to the tentative identification of twenty-four metabolites with matching library scores ranging from 57.7% to 100%, and a mass error ranging from −162.43 to 49.72 ppm (Table 2). These phytochemicals, however, need further elucidation concerning their correct elemental formulas, structures, substructures, and compound classes to acquire higher confidence levels in metabolite annotation as described by metabolomics initiative guidelines [11,31]. Having been subjected to MetFrag (https://msbi.ipb-halle.de/MetFrag/) and Sirus (v.5.8.6), the data revealed significant insights into the metabolite profiles. In this case, 19 out of twenty-four metabolites could be ranked as the top candidates (1st and 2nd) among several potential structures retrieved, affirming the robustness of the results. Meanwhile, the five remaining metabolites, including *m/z* 137.0247 [M-H]⁻, 133.0151 [M-H]⁻, 115.0042 [M-H]⁻ 227.2018 [M + H]⁺, 339.1803 [M + H]⁺, and 171.1387 [M + H]⁺, need to be excluded as they cannot be annotated by either MetFrag or Sirus. Despite the fact that negative ion mode analysis is more suitable for detecting phenolic species, switching to positive ion mode helped confirm the re-existence of quinic acid (*m/z* 193.0729), ethyl gallate (*m/z* 199.0621), and myricitrin (*m/z* 465.1061), while the putative peak germacrone, characterized by *m/z* 219.1393 (Rt = 8.57 min), was also detected. Akin to CANOPUS, the annotated metabolites were classified into five groups: simple phenolics (with derivatives), flavonoids, carboxylic acids, sugar alcohols, and terpenes. This supports our proposition that the *S. terebinthifolia* hydroethanolic plant extract is presumably a promising source of antiviral agents, as many of these compounds have previously shown

**Table 1. Antioxidant activity of *S. terebinthifolia* alcoholic extract.**

| Sample | (DPPH assay) $IC_{50}$ µg/mL |
|---|---|
| *S. terebinthifolia* hydroethanolic extract | 16.33 |
| Quercetin | 12.72 |
| Gallic acid | 3.06 |
| Total phenolic content (TPC) | |
| *S. terebinthifolia* hydroethanolic extract | 264.11 mg (GAE)/g DE |

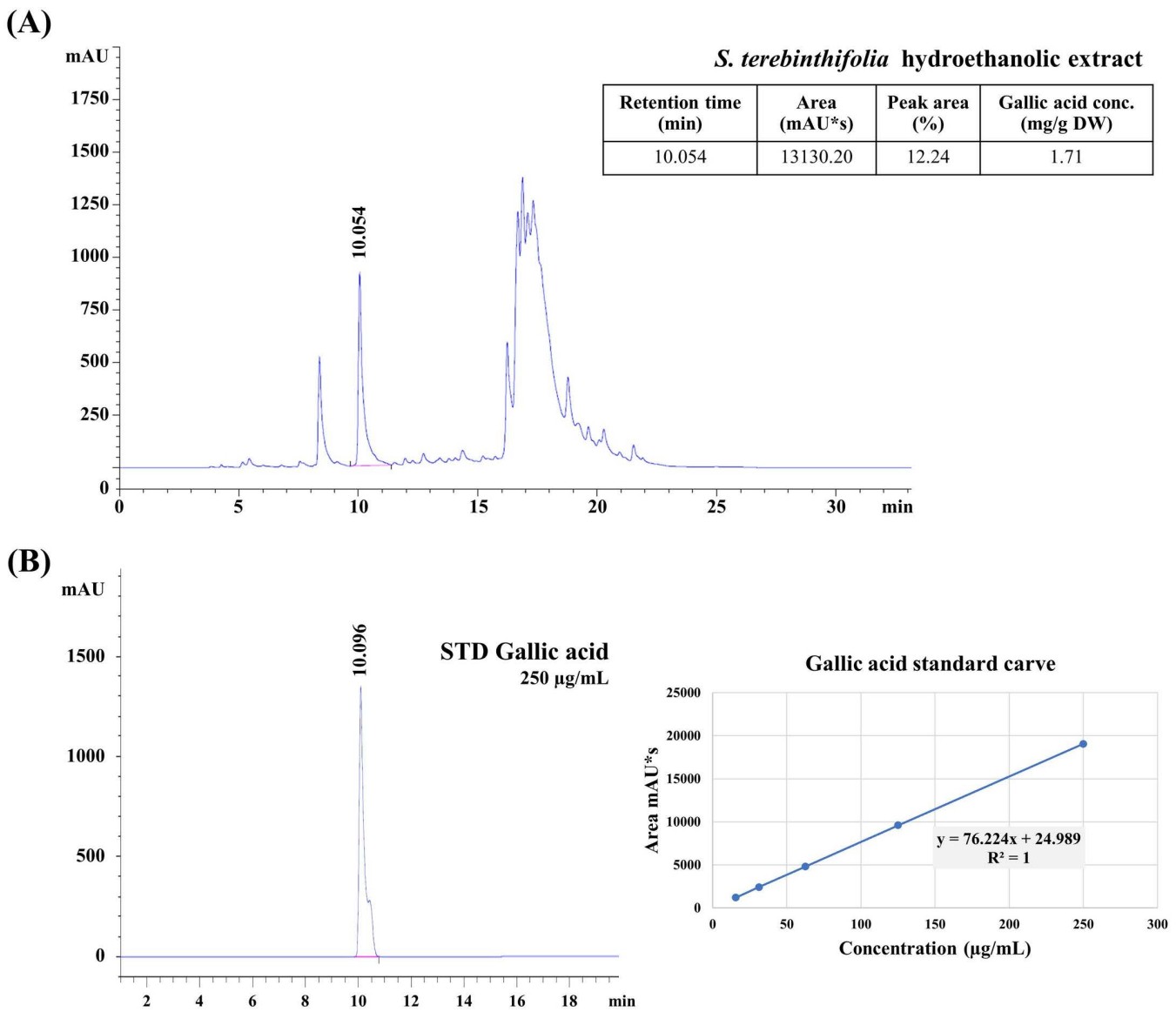

**Fig 1. HPLC analysis of gallic acid as representative marker in *S. terebinthifolia* hydroethanolic extract.** (A) 30% hydroethanolic extract. (B) standard gallic (250 μg/mL) acid and gallic acid standard curve.

anti-influenza activity, particularly shikimate- and phenylpropanoid-related substances [32,33], whose structural annotations are described below.

**3.3.1. Simple phenolics.** As described by CANOPUS, there were eight metabolites that annotated to be solely originated from shikimate-and phenylpropanoid pathways. These included catechol ($m/z$ 109.0317), 3-dehydroshikimic acid ($m/z$ 171.0307), pyrogallol ($m/z$ 125.0270), quinic acid ($m/z$ 191.0651/193.0729), shikimic acid ($m/z$ 173.0485), gallic acid ($m/z$ 169.0178), methyl gallate ($m/z$ 183.0310), ethyl gallate ($m/z$ 197.0522/199.0621), and chlorogenic acid ($m/z$ 353.0910). This result extends the findings of the recently reviewed article [34], which showed that these metabolites share the shikimic route, thus validating the metabolite annotation, as they are clearly relevant to each other from a biosynthetic point of view. High-resolution isotope pattern analysis also revealed that they all received a Sirius score of

**Table 2. Tentative identification and ranked annotation of selected metabolites belonged to *S. terebinthifolia* hydroethanolic extract.**

| No. | UPLC-ESI(±)-QTOF-MS/MS | | | | | | | | MetFrag Rank/DB (F1 Score) | SIRIUS Rank/DB (CSI:FingerID Matching Score%)/Training (YES/NO)§ | SwissADME Lipinski's rule[1]/ Log Po/w (iLOGP)[1] |
|---|---|---|---|---|---|---|---|---|---|---|---|
| | RT (min) | Proposed Structure (% Relative peak area)[2] | Neutral formula (Sirius Score %)* | Neutral mass*** | Theoretical (m/z) [M-H]/ [M+H]+† | Exp. mass (m/z) [M-H]- | Diff Mass (ppm)‡ | Library Score (%)¶ | | | |
| 1 | 0.95 | 4-Hydroxybenzoic acid (2.28) | $C_7H_6O_3$ | 138.03 | 137.0239 | 137.0247 | 5.84 | 100.0 | 2nd/KEGG (0.9759) | ND. | Yes; 0 violation / 0.85 |
| 2 | 3.01 | Pyrocatechol (0.11) | $C_6H_6O_2$ | 110.04 | 109.0290 | 109.0317 | 24.76 | 57.7 | 2nd/KEGG (1.0) | 1st/All databases (100)/ (Yes) | Yes; 0 violation / 1.65 |
| 3 | 2.16 | Gallic acid (8.99) | $C_7H_6O_5$ | 170.02 | 169.0137 | 169.0178 | 24.26 | 93.3 | 2nd/KEGG (0.9731) | 1st/All databases (100)/ (Yes) | Yes; 0 violation / 0.21 |
| 4 | 0.84 | Malic Acid (0.71) | $C_4H_6O_5$ | 134.02 | 133.0137 | 133.0151 | 10.53 | 99.8 | ND. | ND. | Yes; 0 violation / −0.01 |
| 5 | 0.67 | Myo-inositol [M+FA-H]– (1.61) | $C_6H_{12}O_6$ | 180.06 | 225.0610 | 225.0624 | 6.22 | 94.4 | 9th/ NORMAN (0.3364) | 1st/All databases (100)/ (Yes) | Yes; 1 violation: NHorOH > 5 / 0.31 |
| 6 | 2.16 | Pyrogallol (1.88) | $C_6H_6O_3$ | 126.03 | 125.0239 | 125.0270 | 24.80 | 98.2 | 3th/ KEGG (0.7832) | 1st/All databases (90.09)/ (Yes) | Yes; 0 violation / 0.97 |
| 7 | 6.59 | Azelaic Acid (3.24) | $C_9H_{16}O_4$ | 188.10 | 187.0970 | 187.0993 | 12.30 | 91.8 | 1st/ KEGG (1.0) | 1st/All databases (100)/ (Yes) | Yes; 0 violation / 1.44 |
| 8 | 0.76 /0.76 | Quinic acid (7.61) | $C_7H_{12}O_6$ | 192.06 | 191.0556 193.0712 [M+H]+ | 191.0651 193.0729 [M+H]+ | 49.72 /8.81 | 94.6/83.0 | 1st/ KEGG (1.0) | 1st/All databases (99.36)/ (Yes) Negative 1st/Plantcyc 1st/Natural Products (58.90)/ (Yes) positive | Yes; 0 violation / −0.12 |
| 9 | 3.91 | Methyl gallate (0.13) | $C_8H_8O_5$ | 184.04 | 183.0294 | 183.0310 | 8.74 | 54.4 | 2nd/ NORMAN (1.0) | 3rd/All databases, 2nd/KEGG Mine (83.25)/ (Yes) | Yes; 0 violation / 0.97 |
| 10 | 0.91 | Shikimic acid (23.99) | $C_7H_{10}O_5$ | 174.052824 | 173.0450 | 173.0485 | 20.23 | 89.5 | 4th/ KEGG (0.6808) | 1st/All databases (100)/ (Yes) | Yes; 0 violation / 0.45 |
| 11 | 5.26 | Suberic acid (2.75) | $C_8H_{14}O_4$ | 174.09 | 173.0814 | 173.0825 | 6.36 | 89.7 | 2nd/KEGG (1.0) | 1st/All databases (100)/ (Yes) | Yes; 0 violation / 1.15 |
| 12 | 12.40 | Hexadecanedioic acid (0.59) | $C_{16}H_{30}O_4$ | 286.21 | 285.2066 | 285.2084 | 6.31 | 100 | 2nd/KEGG (1.0) | 1st/All databases (100)/ (Yes) | Yes; 0 violation / 2.98 |
| 13 | 9.78 | Dodecanedioic acid (1.20) | $C_{12}H_{22}O_4$ | 230.15 | 229.1440 | 229.1463 | 10.04 | 67.7 | 1st/ KEGG (1.0) | 1st/All databases (100)/ (Yes) | Yes; 0 violation / 1.83 |
| 14 | 5.22 /5.21 | Ethyl gallate (40.20) | $C_9H_{10}O_5$ | 198.05 | 197.0450 199.0607 [M+H]+ | 197.0522 199.0621 [M+H]+ | 36.54 7.03 | 96.8/65.4 | 3th/ NORMAN (0.9467) | 1st/All databases (98.51)/ (Yes) Negative 1st/All databases (98)/ (Yes) Positive | Yes; 0 violation / 1.21 |
| 15 | 8.75 | Undecanedioic acid (1.22) | $C_{11}H_{20}O_4$ | 216.14 | 215.1283 | 215.1303 | 9.30 | 93.1 | 1st/ NORMAN (1.0) | 1st/All databases (100)/ (Yes) | Yes; 0 violation / 1.92 |
| 16 | 0.84 | 3-Dehydroshikimic acid (0.21) | $C_7H_8O_5$ | 172.04 | 171.0294 | 171.0307 | 7.60 | 94.3 | 7th/KEGG (0.8957) | 1st/All databases (100)/ (Yes) | Yes; 0 violation / 0.25 |
| 17 | 0.84 | Maleic Acid (0.71) | $C_4H_4O_4$ | 116.01 | 115.0031 | 115.0042 | 9.56 | 99.6 | ND. | ND. | Yes; 0 violation / −0.01 |

*(Continued)*

| No. | UPLC-ESI(±)-QTOF-MS/MS | | | | | | | | MetFrag Rank/DB (F1 Score) | SIRIUS Rank/DB (CSI:FingerID Match-ing Score%) /Training (YES/NO)§ | SwissADME Lipinski's rule[1]/ Log Po/w (iLOGP)[1] |
|---|---|---|---|---|---|---|---|---|---|---|---|
| | RT (min) | Proposed Structure (% Relative peak area)[2] | Neutral formula (Sirius Score %)* | Neutral mass*** | Theoretical (m/z) [M-H]/ [M+H]+† | Exp. mass (m/z) [M-H]- | Diff Mass (ppm)‡ | Library Score (%)¶ | | | |
| 18 | 3.79 | Chlorogenic acid (0.57) | $C_{16}H_{18}O_9$ | 354.10 | 353.0873 | 353.0910 | 10.48 | 100.0 | 1st/ KEGG (1.0) | ND. | Yes; 1 violation: NHorOH > 5 / 0.96 |
| 19 | 5.60/ 5.60 | Myricitrin (0.82) | $C_{21}H_{20}O_{12}$ | 464.10 | 463.0877 465.1033 [M+H]+ | 463.1002 465.1061 [M+H]+ | 26.99 6.02 | 97.0/89.4 | 1st/ KEGG (1.0) | 1st/All databases (100)/ (Yes) Negative 1st/All databases (100)/ (Yes) Positive | No; 2 violations: NorO > 10, NHorOH > 5 Ghose (Yes)[3] / 0.92 |
| 20 | 6.85 | Afzelin (0.25) | $C_{21}H_{20}O_{10}$ | 432.11 | 431.0978 | 431.1052 | 17.17 | 97.9 | 1st/ KEGG (1.0) | 1st/All databases (100)/ (Yes) | Yes; 1 violation: NHorOH > 5 / 1.84 |
| 21 | 10.47 | Xanthotoxol gera-nyl ether[4] | $C_{21}H_{22}O_4$ | 338.15 | 339.1596 | 339.1803 | 61.03 | 75.6 | ND. | ND. | Yes; 0 violation / 3.88 |
| 22 | 8.57 | Germacrone[4] | $C_{15}H_{22}O$ | 218.17 | 219.1749 | 219.1393 | −162.43 | 95.9 | ND. | 1st/training set (67.69) | Yes; 0 violation / 2.96 |
| 23 | 10.68 | (S)-citronellic acid[4] | $C_{10}H_{18}O_2$ | 170.13 | 171.1385 | 171.1387 | 1.17 | 84.7 | 2nd/KEGG (0.926) | ND. | Yes; 0 violation / 2.28 |
| 24 | 11.86 | Myristoleic acid[4] | $C_{14}H_{26}O_2$ | 226.19 | 227.2011 | 227.2018 | 3.08 | 88.3 | – | ND. | Yes; 0 violation / 3.39 |

*Based on both high-resolution isotope pattern analysis (as defined by Sirius) and the MetFrag web service.

**Formula annotation is based on MetFrag web service.

***Annotated by MetFrag web service.

†Calculated by using a web service (https://www.sisweb.com/referenc/tools/exactmass.htm).

‡Calculated by a mass error calculation tool (https://warwick.ac.uk/fac/sci/chemistry/research/barrow/barrowgroup/calculators/mass_errors/).

§Following the CSI:FingerID structural training set tandem with SIRIUS, available at https://www.csi-fingerid.uni-jena.de/v2.6/api/fingerid/trainingstruc-tures?predictor=2).

¶Matching against tandem mass spectral data was derived from the Natural Products HR-MS/MS Library (version 2.0) and NIST 2017 MS/MS library. Metabolites denoted as "ND. were perceived as having an "improper structure".

Metabolites No. 8, 14 and 19 were simultaneously detected both negative and positive ion modes.

[1]Drug likeliness and lipophilicity of the detected metabolites were assessed using Swiss ADME (http://www.swissadme.ch/index.php) to estimate their potential for penetrating viral envelopes and targeting the cap-binding domain (PB2; 4NCE) based on the iLOGP value. Metabolites were categorized as follows: Highly Hydrophilic: Log Po/w (iLOGP) < 0, suggesting a stronger affinity for water compared to octanol. Moderately Hydrophilic: $0 \leq$ Log Po/w (iLOGP) < 1, indicating a balanced affinity for both octanol and water, which implies moderate hydrophobicity and potential for diffusing through viral envelopes. Highly Hydrophobic: Log Po/w (iLOGP) $\geq 3$, reflecting a strong preference for octanol over water, and the greatest potential for crossing viral envelopes.

[2]Calculated from the total metabolites selected from negative ion mode analysis.

[3]Myricitrin satisfies Ghose's rule.

[4]The % relative peak area for metabolites 21–24, specifically detected in positive mode, was not calculated.

100%, indicating their correct molecular formulas were determined (Table 2). Besides the different ranks assigned by MetFrag (1st, 2nd, 3rd, and 7th), CSI:FingerID supports greater structural reliability, re-ranking them as the top candidates (1st and 2nd) among the hundred potential structures postulated, presumably due to their pre-existence in the training dataset (https://www.csi-fingerid.uni-jena.de/v2.6/api/fingerid/trainingstructures?predictor=2). CSI:FingerID also showed

that many substructures (finger prints) representing basic ring structures (e.g., $C_6$-$C_1$ and $C_6$), hydroxy (-OH), carboxylic acid (-COOH), methoxy (-OCH$_3$), and ethoxy (-OC$_2$H$_5$) groups were either widely or individually detected in the query subjects. Methyl- and ethyl gallate share similar basic ring features ($C_6$-$C_1$) with their chemical parent, gallic acid, characterized by substructures encoded by "ECFP6:-169", "c(:c:c(:c:c1)~[!#1]):c1[CH0](~[!#1]~[!#1]", and "Oc1ccc(C=O) cc10". The former, however, exclusively contains molecular fingerprints that illustrate the methoxy group ([!#6,!#1]~[CH3]; 79%) and successive ether bond formation ([!#1]O[CH3]; 77%) (Fig 2). This was corroborated by the detected product ion at $m/z$ 124.0175 (base peak), illustrating the pyrogallol moiety ($C_6H_4O_3$•-). Moreover, its fragmentation pattern closely matched that of authentic methyl gallate in the database (https://pubchem.ncbi.nlm.nih.gov/image/ms.cgi?pea ks=124.02058:100,183.03558:37.65,168.00569:20.51,78.01257:19.89,123.00983:8.58). Besides the gallic acid-derived moiety (ECFP6: −1691511413; 100%), substructures representing a carboxyl ester (CX3;$([RO][#6]),$(H1R0)](=[OX1] [OX2][#6;!$(C=[O,N,S]; 100%), and one indicating successful ether-bond formation ([!#1][CH2]OC(=O)[!#1]; 100%) were detected in the latter query subject. Its MS/MS spectrum also included daughter ions at $m/z$ 169.0155 and $m/z$ 124.0179 (base peak), representing the presence of an ethyl group and pyrogallol, respectively [35]. Despite the biogenesis of methyl and ethyl gallate has not been reported yet, CANOPUS suggested that both metabolites belong to the main class "Galloyl esters", describing the ester derivatives of 3,4,5-trihydroxybenzoic acid.

Similarly, 3-Dehydroshikimic acid and shikimic acid shared various substructures such as "CCCCCC(=O)O; 93%," "CCCCCCC; 90%," and "C1CCC=CC1; 89%" as the former is a product of shikimate dehydrogenase [36]. Substructures that indicated successful dehydrogenation of the hydroxy group, such as "#6(-,:[#6])(+,:[#8]) (C(-C)(-C)(=O); 94%)," "O=C-1CCCC=C1; 84%," and "CC(=O)CO; 87%" were exclusively detected in this query subject as a result (Fig 3). Catechol, pyrogallol, and quinic acid have been proposed to be involved with gallic acid, which previously ranked as the top candidate (1st), as they share the shikimate route. Therefore, their successful annotation is reliable because they share identical synthetic routes in various plant species [37,38] and might be strengthened by the existing data in the CSI: FingerID training set (https://www.csi-fingerid.uni-jena.de/v2.6/api/fingerid/trainingstructures?predictor=2) (Fig 4).

Despite the negative ion with $m/z$ 353.0910 not being annotated as 'chlorogenic acid' by CSI:FingerID, as it was previously in MetFrag with a perfect match (100% hit score library), it has the correct neutral formula of $C_{16}H_{18}O_9$ (Sirius score 100%), which is consistent with the authentic compound in the database (https://pubchem.ncbi.nlm.nih.gov/com-pound/Chlorogenic-Acid). This incomplete annotation is presumably due to the insufficient fragmentation spectra of this query metabolite, as Sirius prefers rich fragmentation spectra (up to 60 peaks) and automatically determines which peaks should be annotated or excluded, as outlined in the Sirius user manual (https://bio.informatik.uni-jena.de/repository/dist/ de/unijena/bioinf/ms/sirius/4.0.1/). While further elucidation (e.g., optimizing collision-induced energy) is required, the fragmented ion with $m/z$ 191.0575 (base peak, 100% relative intensity) observed in the MS/MS spectrum indicates the complete loss of quinic acid from the caffeic moiety, consistent with its authentic structure in PubChem (https://pubchem. ncbi.nlm.nih.gov/compound/Chlorogenic-Acid#section=Computed-Properties) (Fig 5). Accordingly, they are inferred to be the same substance.

**3.3.2. Flavonoids.** Myricitrin and afzelin, two glycosylated flavonoids detected in the *S. terebinthifolia* hydroethanolic extract, were predicted to originate from the phenylpropanoid pathway. According to the Sirius and CSI:FingerID web services, the former, simultaneously detected in negative and positive modes, is best described in terms of its molecular formula and structural data (Fig 6). Substructure detection clearly showed an existing of basic flavonoid ring system, characterized by C6(ring A)-C3(ring C)–C6(ring B), and rhamnopyranoside moiety present in this query subject. Substructures encoded by "c(:c (:c (:c (:c(:c:c(:c:c1)~[#8])~[#8]):c:1:2)~[#8][#8]):o:2; 100%" and "[#8]=,:[#6]-,[#6]-,:[#6]- ,:[#6]-,:[#6](-,:[#8])-,:[#6] (O=C-C-C-C-C(O)-C); 100%" indicating the correlation between rings A and C systems, thought to synthesize from the Claisen condensation of *p*-coumaroyl-CoA and three malonyl-CoA units catalyzed by chalcone synthase [39]. Meanwhile, two fingerprints illustrating the existence of a basic ring B system ("Oc1cccc(O)c10; 94%") and rhamnopyranoside moiety ("ECFP6:924408534; 94%") were clearly detected in the same potential candidate ($C_{21}H_{20}O_{12}$).

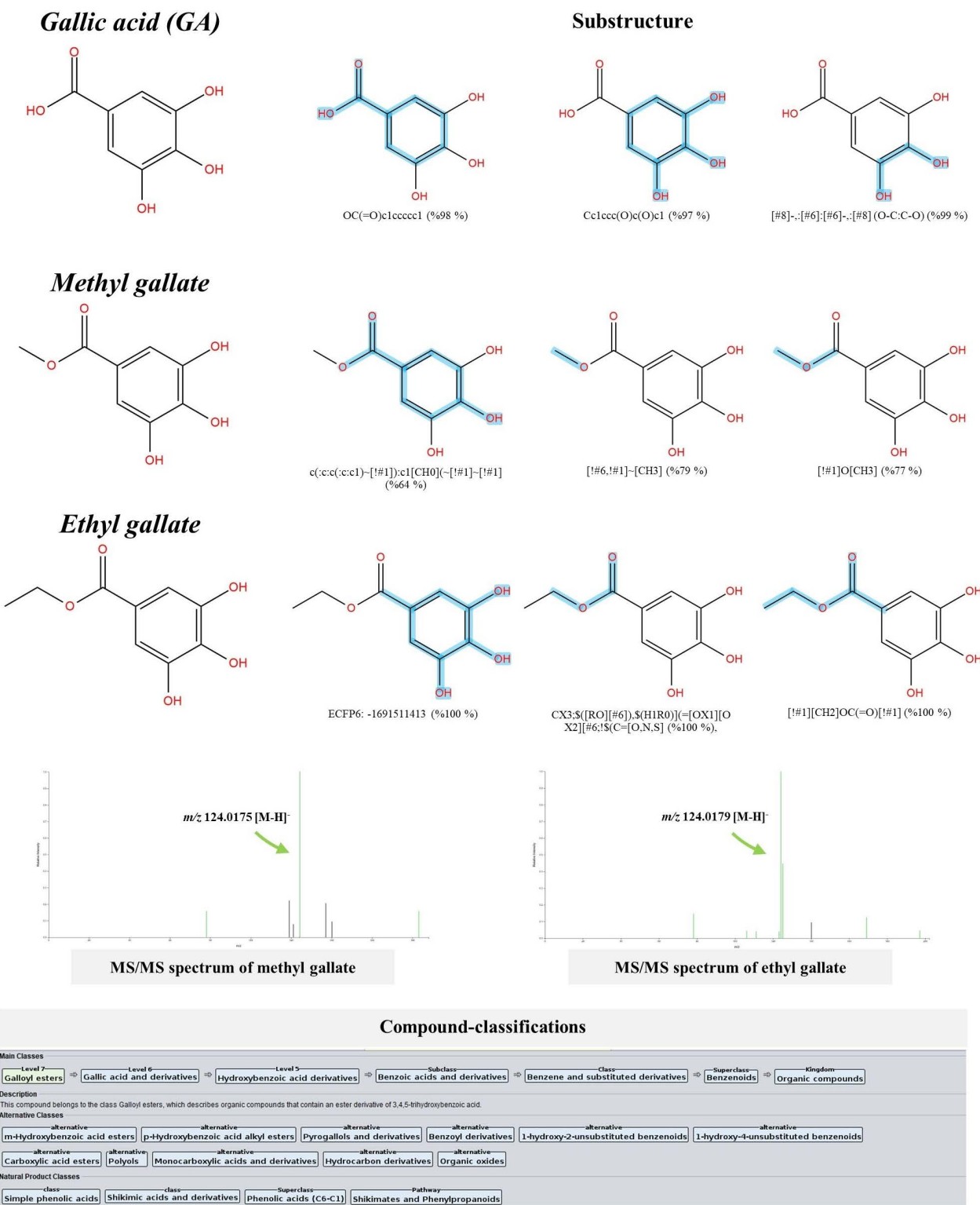

**Fig 2. Structural annotation of putative gallic acid, methyl gallate and ethyl gallate detected from the _S. terebinthifolia_ hydroethanolic extract.**

**Fig 3. Structural annotation of putative shikimic acid and 3-Dehydroshikimic acid detected from the *S. terebinthifolia* hydroethanolic extract.**

This is evidenced by identifying the fragment ion at *m/z* 316.02340 (base peak in negative mode) and *m/z* 319.0499 (in positive mode; 1.0 relative intensity) in its MS/MS spectrum, signifying the complete loss of the alpha-L-rhamnopyranosyl residue from myricetin (as the aglycone core structure). A similar observation was noted with putative afzelin (*m/z* 431.1052; $C_{21}H_{20}O_{10}$; Rt = 6.85 min), where molecular fingerprints conforming to aglycone and glycone moieties were found in this query subject. Indicated by the substructures 'c(:c(:c:c:c:c:1):c:1:2)(:c:2~[#8]~[#8]; 100%)' and 'c1ccccc1; 100%', the previous shows the successive Claisen condensation between rings A and C, and the latter represents the B ring system (derived from *p*-coumaroyl-CoA), collectively signifying the presence of a kaempferol core structure. CSI:FingerID also detected the rhamnosyl moiety and its successive O-glycosylation with the C ring system, symbolized by '[!#1][CH]1OCHCHCH[CH]1[OH]; 99%' and '[CH1][OH0]c; 99%' in the same structural candidate. The observed product ion at *m/z* 284.0341 [M-H]⁻ (base peak; 1.0 relative intensity), indicating the complete loss of the rhamnosyl unit from kaempferol, strengthens the annotation of this query subject (https://pubchem.ncbi.nlm.nih.gov/compound/Afzelin#section=Other-MS).

**3.3.3. Carboxylic acids, myoinositol, and germacrone.** Multiple forms of dicarboxylic acids were also detected in the *S. terebinthifolia* hydroethanolic extract, including suberic acid ($C_8H_{14}O_4$), azelaic acid ($C_9H_{16}O_4$), undecanedioic acid ($C_{11}H_{20}O_4$), dodecanedioic acid ($C_{12}H_{22}O_4$), and hexadecanedioic acid ($C_{16}H_{30}O_4$). According to CSI:FingerID webservice, they share many identical features but differ from one another in their specific chain lengths (Fig 7). Taking putative hexadecanedioic acid (*m/z* 285.2084; Rt = 12.40 min) as an example, its accuracy annotation was improved from the second to the best-ranked candidate (1st). As denoted by '[!#1][CH2][CH2]C(=O)[!#1]; 100%' and 'CCCCCCCC=O; 93%',

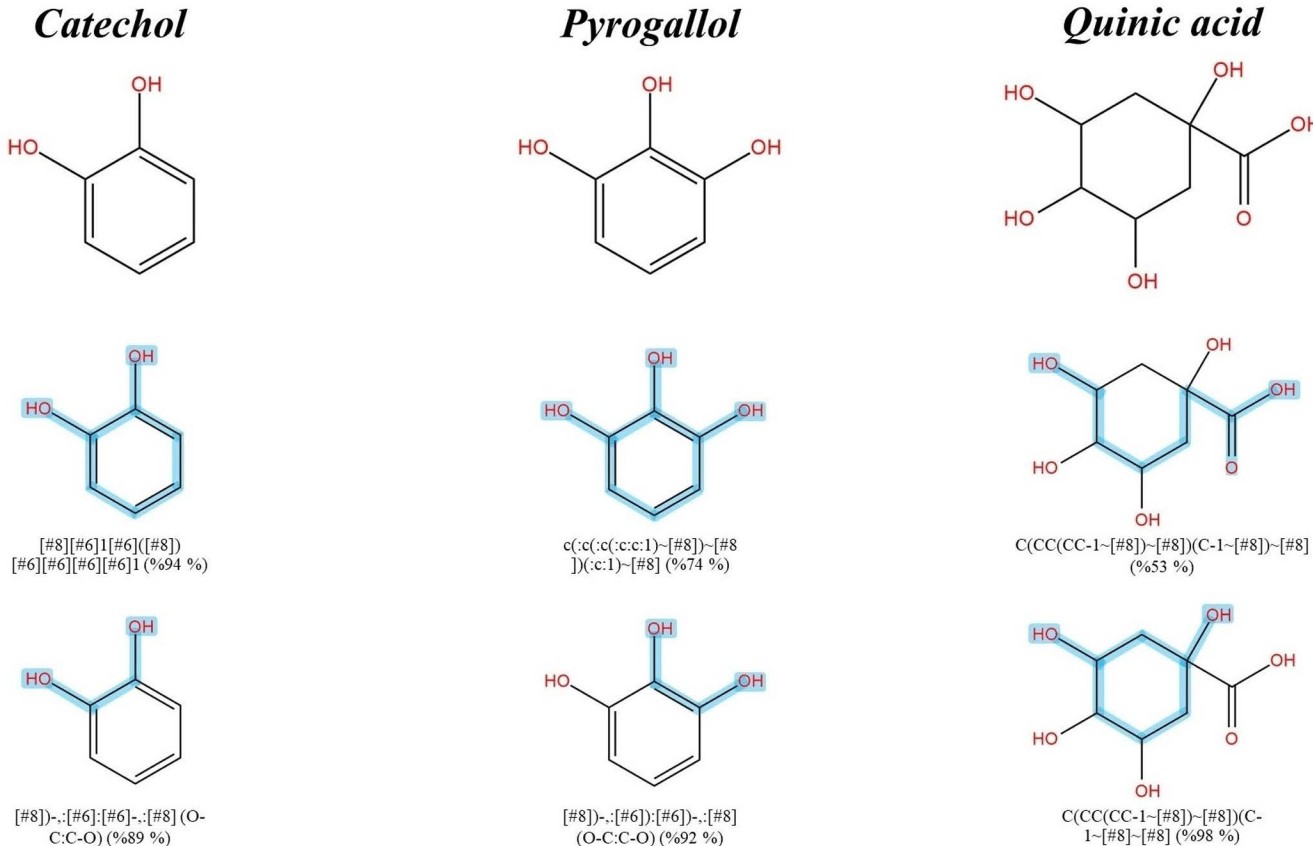

**Fig 4. Structural annotation of putative catechol, pyrogallol and quinic acid detected from the *S. terebinthifolia* hydroethanolic extract.**

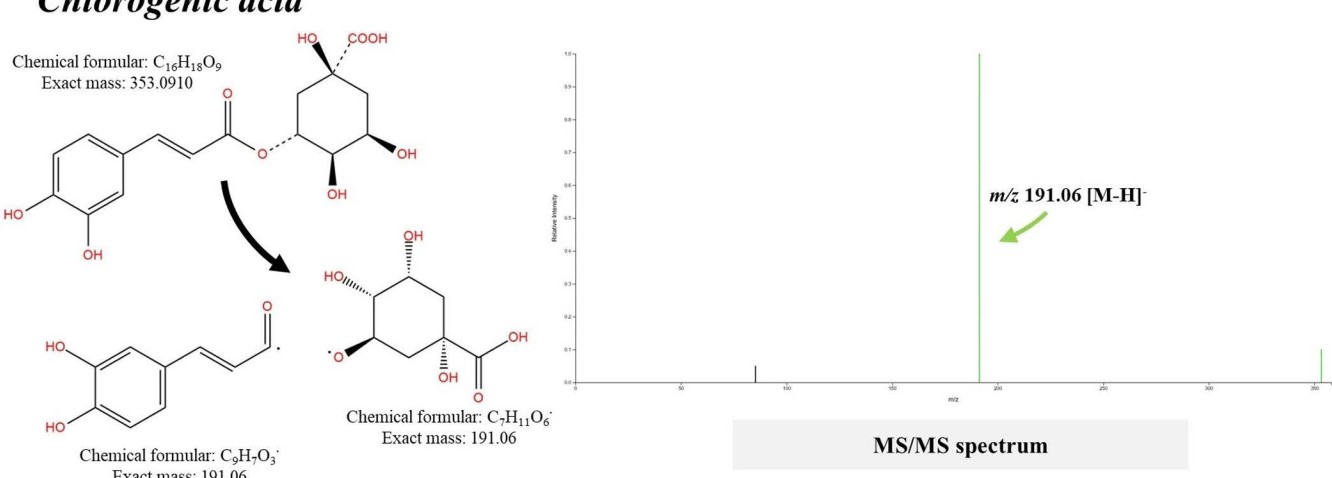

**Fig 5. Structural annotation of putative chlorogenic acid (m/z 353.0910 [M-H]⁻) detected from the *S. terebinthifolia* hydroethanolic extract.**

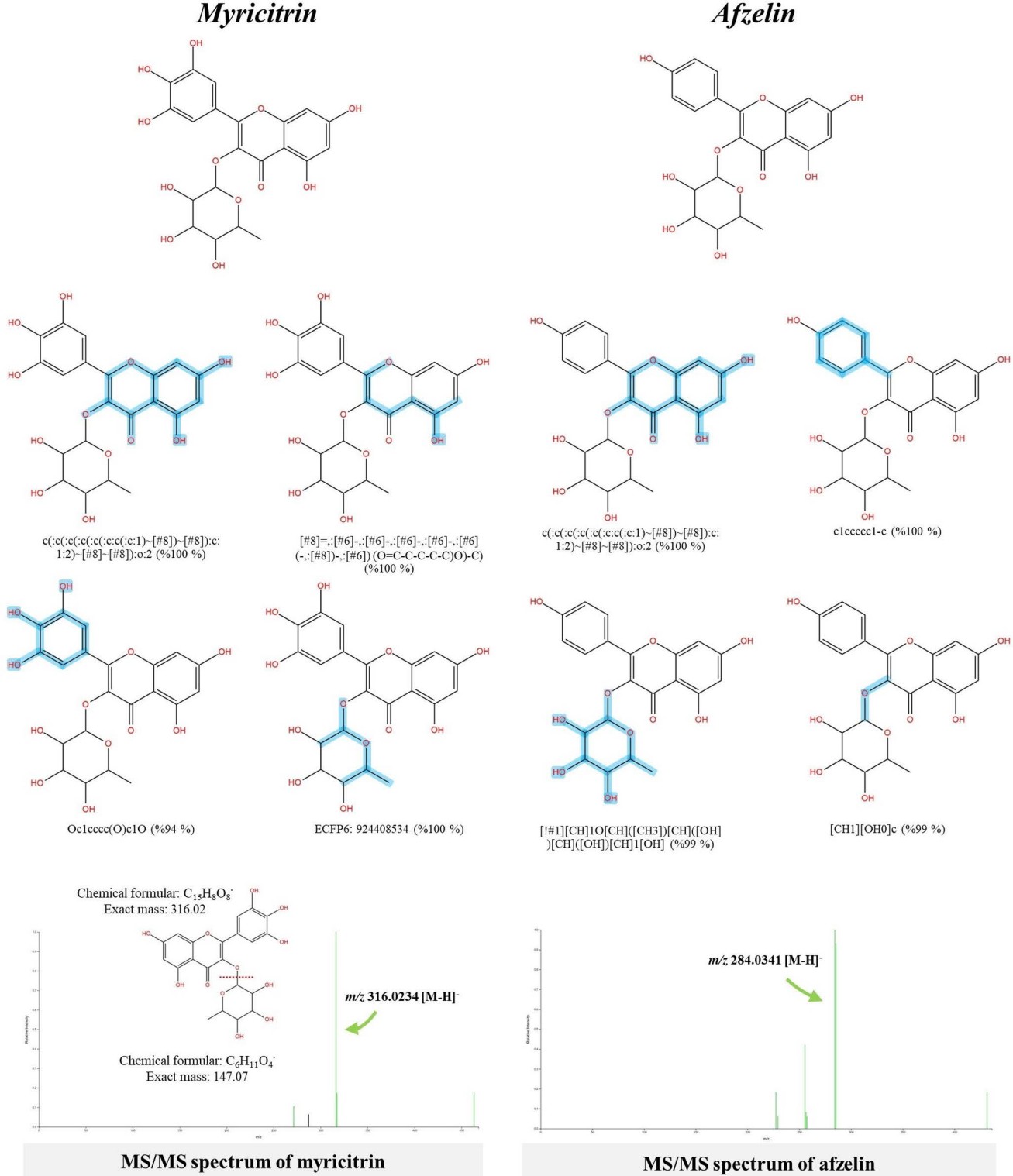

**Fig 6. Structural annotation of putative myricitrin and afzelin detected from the *S. terebinthifolia* hydroethanolic extract.**

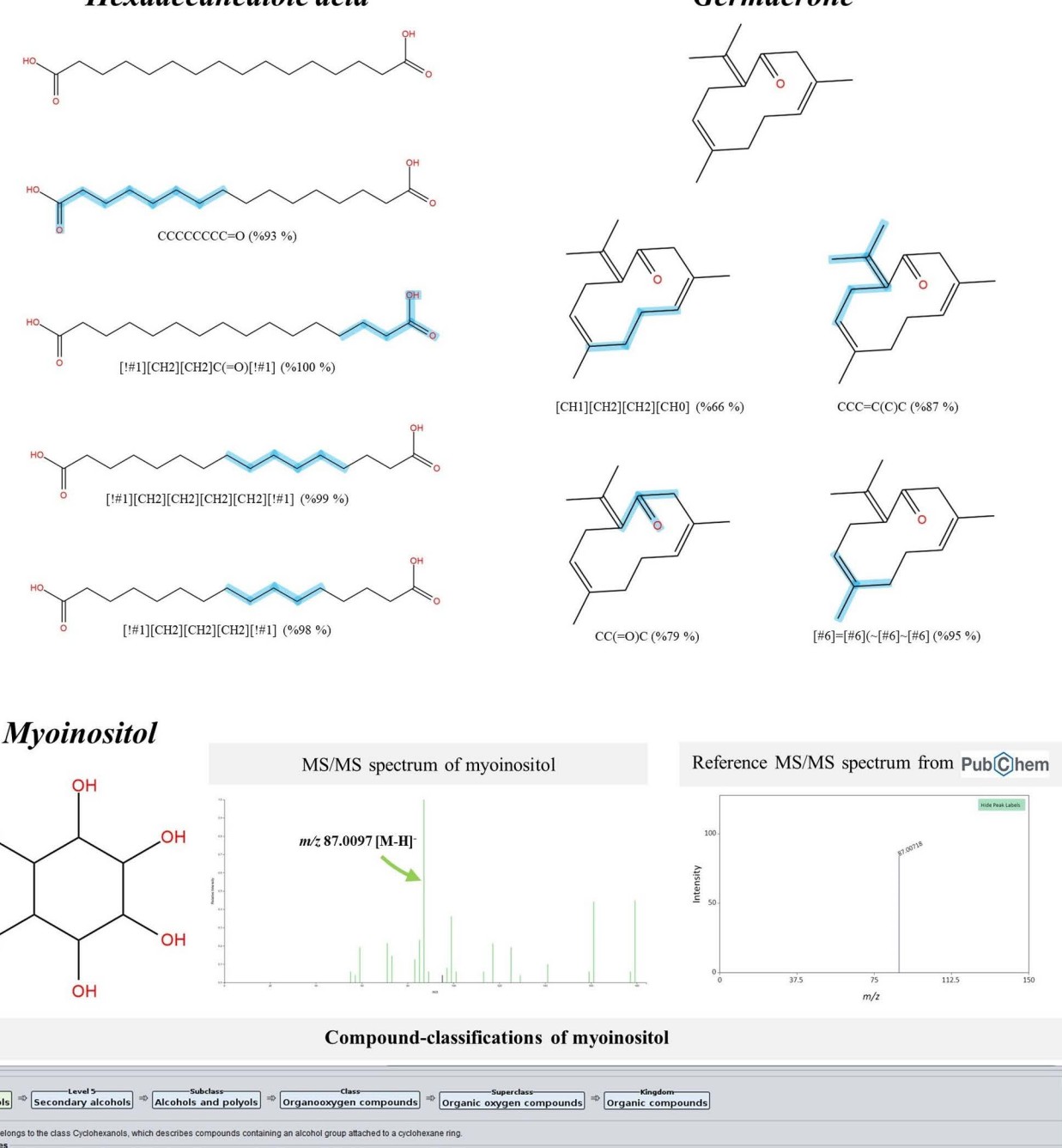

**Fig 7. Structural annotation of putative hexadecanedioic acid, germacrone and myoinositol detected from the *S. terebinthifolia* hydroethanolic extract.**

these substructures illustrate the atomic correlation between aliphatic hydrocarbons ($C_3$-$C_7$) and a carboxyl group detected at both ends. Meanwhile, the alkyl substructures, encoded by '[!#1][CH2][CH2][CH2][CH2][!#1]; 99%' and '[!#1][CH2][CH2][CH2][!#1]; 98%', were deduced in the same query subject. Similarly, the rank of putative myoinositol ($m/z$ 225.0624) shifted from the 9th to the 1st candidate, presumably because this metabolite was included in the training set (https://www.csi-fingerid.uni-jena.de/v2.6/api/fingerid/trainingstructures?predictor=2). Although the putative germacrone ($m/z$ 219.1393 [M + H]+; Rt = 8.57 min) could not be further annotated by MetFrag, its rank shifted entirely to the top candidate (1st) upon aligning with the CSI:FingerID training set. Notably, CSI-FingerID facilitated the comprehensive mapping of substructures that depicted the stepwise cyclization of three C5 isoprene units into a cyclic sesquiterpene, including '[CH1][CH2][CH2][CH0]; 66%', 'CCC=C(C)C; 87%', 'CC(=O)C; 79%' and '[#6]=[#6](~[#6]~[#6]; 95%'. CANOPUS has confirmed that this metabolite belongs to the class of sesquiterpenoids (superclass), which are terpenes with three consecutive isoprene units, thereby unraveling its potential biosynthetic origin (Fig 7).

### 3.4. Anti-viral and Hemolytic Activities of the plant extracts

The cytotoxicity assays against MDCK cells showed that extract concentrations were non-toxic with a $CC_{50}$ above 50 µg/mL. After dilution, the extracts were mixed with the virus and applied to MDCK cells for infection, followed by a 24-hour incubation to assess antiviral activity. The *S. terebinthifolia* hydroethanolic extract had an $IC_{50}$ of 2.21 µg/mL against the influenza A/PR/8/34 virus, about 1.96 times more effective than gallic acid ($IC_{50}$ = 4.35 µg/mL) (Fig 8, Table 3). Preliminary calculations revealed that the gallic acid content in the extract was 0.27 µg/mL, indicating that its increase did not correspond with enhanced antiviral activity. This suggests that other phytochemicals may contribute to a synergistic effect in this circumstance. The extract's presence during multiple viral life cycle stages suggests its inhibitory effect. The significant difference between $IC_{50}$ and $CC_{50}$ values shows effective antiviral concentration without harming MDCK cells. Additionally, the extract showed no hemolytic activity towards hRBCs, even at 2,000 µg/mL, indicating its potential as a safe antiviral solution.

### 3.5. Virtual screening for potential anti-influenza agents

Computer-aided screening indicates that the docked metabolite presents in the *S. terebinthifolia* extract showed a superior interaction with the catalytic site of NA compared to HA, suggesting that HA may be an off-target as reported herein (Table 4). Previous studies have also demonstrated that metabolites originating from the shikimate and phenylpropanoid biosynthetic pathways (e.g., gallic acid, quercetin, and various glycosylated analogs) can specifically interact with the cap-binding domain (PB2) of RNA polymerase by forming hydrophobic interactions with highly conserved residues such as Phe323, His357, Phe363, and Phe404, thereby disrupting the replication processes of various influenza viruses [40,41]. Here, in silico screening also revealed that several metabolites, such as myricitrin, afzelin, chlorogenic acid, and methyl- and ethyl-gallate, exhibit promising binding properties toward PB2, similar to those observed with the reference ligands such as gallic acid, quercetin, and hyperoside. Correspondingly, NA and PB2 are likely targeted in the underlying anti-influenza activity of the hydroethanolic plant extract, with the molecular mechanisms of action explained below.

### 3.6. Molecular docking against neuraminidase (NA)

According to Zhang et al. [24], gallic acid and methyl gallate have demonstrated anti-neuraminidase activity, with $IC_{50}$ values of 450 nM and 490 nM, respectively. Although they are less potent than oseltamivir acid ($IC_{50}$ = 250 nM), their effectiveness remains remarkable. To determine the inhibitory effects of other bioactive constituents, the binding energies of both phenolics can potentially be used as benchmarks. Therefore, the docked metabolites whose binding energies higher or slightly different from 46.69 should be considered as behaviorally active. Also, they are expected to precisely interact with the NA catalytic residues by forming hydrogen bonds and/or electrostatic bonds, with bond distances not exceeding 3.00 Å and 5.00 Å, respectively [44]. Based on these criteria, 12 out of 18 docked metabolites were identified as having

**(A)**

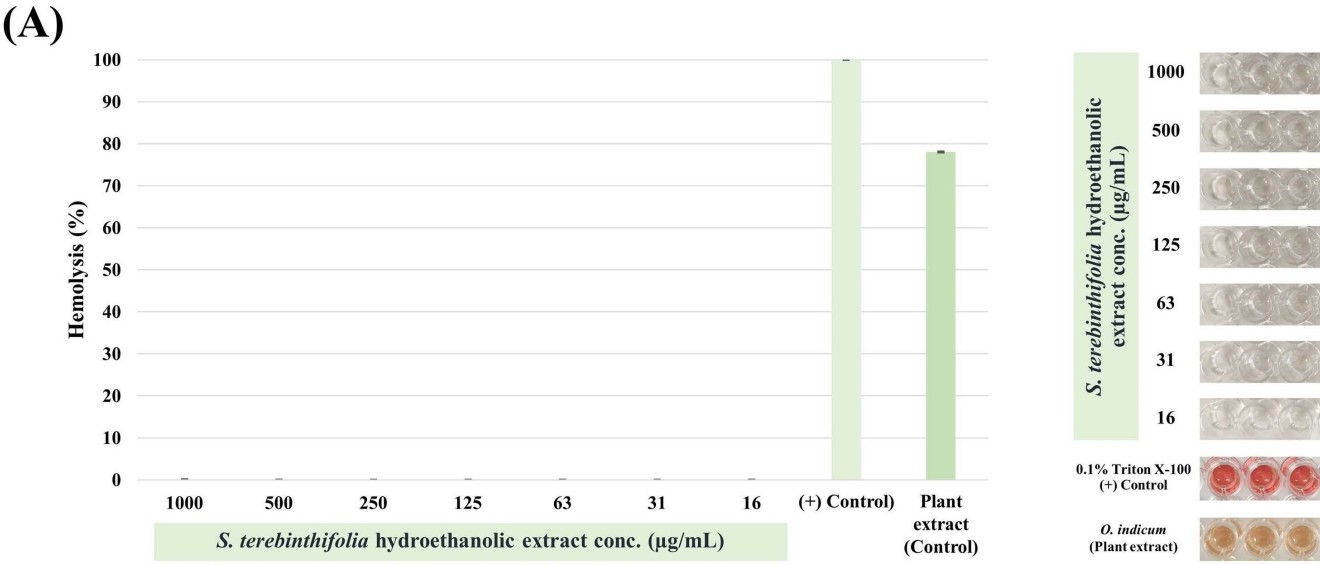

**(B)**

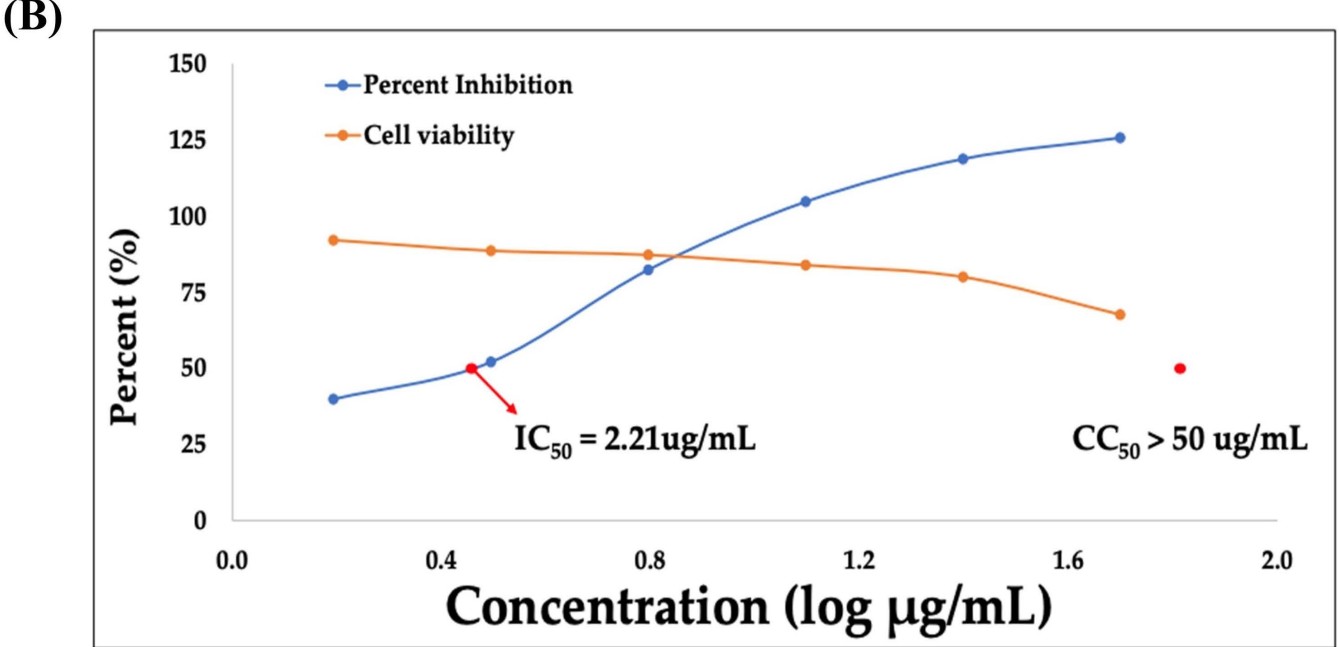

**Fig 8. Antiviral activity of *S. terebinthifolia* hydroethanolic extract, where IC$_{50}$ defines the concentration of a substance that required to inhibit viral replication by 50%.** The lower IC$_{50}$ values signify a greater antiviral potency; (C) CC$_{50}$ is used in the framework of cytotoxicity assays, meaning the concentration of a plant extract and/or the authentic GA (1) that cause a 50% reduction in the viability of the MDCK cells. In this case, the IC$_{50}$ value of *S. terebinthifolia* aqueous ethanol extract is significantly less than its CC$_{50}$ value. This indicated its effective antiviral without inducing any harmful effects on the host cells.

potential anti-influenza activities. Although seven metabolites—catechol, myo-inositol, pyrogallol, quinic acid, shikimic acid, and 3-dehydroshikimic acid—were excluded, they are supposed to play other important roles, such as reducing oxidative stress during viral infections, presumably because of their antioxidant properties [4].

**Table 3. Comparison of anti-influenza and hemolytic activities of *S. terebinthifolia extract*.**

| Sample | Anti-influenza A/Puerto Rico/8/34 (H1N1) | | | Hemolytic activity (µg/mL) |
|---|---|---|---|---|
| | IC$_{50}$ (µg/mL) | Gallic acid** content (µg/mL) | CC$_{50}$ (µg/mL) | |
| *S. terebinthifolia* hydroethanolic extract | 2.21 | 0.27 | >50 | > 2000 |
| Gallic acid* | 4.35 | – | >100 | N.A. |

*Based on our previous describe (Klamrak et. al, 2023). N.A.: Not applicable.

**This value is calculated using this equation: % relative HPLC peak area X IC$_{50}$ value/100.

**Table 4. Target specificity of query metabolites towards HA, NA, and PB2-subunit of influenza A/PR/8/34 virus.**

| Compound | | PubChem CID | Fitness Score (GOLD score) | | |
|---|---|---|---|---|---|
| | | | HA (PDB: 1RU7) | NA (PDB: 6HP0) | PB2 (PDB: 4NCE) |
| *S. terebinthifolia's* metabolites | | | | | |
| 1 | Catechol | 289 | 27.79 | 36.45 | 35.80 |
| 2 | Gallic acid | 370 | 33.45 | 46.69 | 42.71 |
| 3 | Myo-inositol | 892 | 28.44 | 41.01 | 33.86 |
| 4 | Pyrogallol | 1057 | 31.12 | 43.55 | 37.54 |
| 5 | Azelaic Acid | 2266 | 37.70 | 51.68 | 44.98 |
| 6 | Quinic acid | 6508 | 32.55 | 42.06 | 36.57 |
| 7 | Methyl gallate | 7428 | 37.21 | 50.99 | 47.98 |
| 8 | Shikimic acid | 8742 | 32.20 | 41.76 | 41.86 |
| 9 | Suberic acid | 10457 | 35.96 | 46.74 | 42.37 |
| 10 | Hexadecanedioic acid | 10459 | 44.15 | 59.80 | 60.72 |
| 11 | Dodecanedioic acid | 12736 | 41.78 | 58.97 | 54.94 |
| 12 | Ethyl gallate | 13250 | 38.47 | 54.18 | 51.33 |
| 13 | Undecanedioic acid | 15816 | 39.48 | 59.32 | 49.34 |
| 14 | 3-Dehydroshikimate | 439774 | 29.51 | 39.43 | 40.87 |
| 15 | Trans-Chlorogenic Acid (NA inhibitor) | 1794427 | 40.27 | 58.45 | 55.65 |
| 16 | Myricitrin | 5281673 | 57.00 | 69.62 | 69.28 |
| 17 | Afzelin | 5316673 | 55.57 | 68.59 | 69.15 |
| 18 | Germacrone | 6436348 | 28.80 | 43.54 | 41.08 |
| | Control ligand | | | | |
| | Sialic acid | 445063 | 42.45 | 56.45 | – |
| | Isoquercetin* | 5280804 | 51.55 | 74.27 | 70.50 |
| | Rutin* | 5280805 | 56.66 | 85.03 | – |
| | Hyperoside* | 5281643 | 52.44 | 76.61 | 60.92 |
| | Isorhamnetin* | 5281654 | 48.49 | 59.65 | – |
| | Myricetin** | 5281672 | 62.44 | 46.90 | 60.99 |
| | Favipiravir-RTP | 5271809 | – | – | 82.03 |
| | Quercetin* | 5280343 | 47.66 | 57.29 | 59.86 |
| | Quercimeritrin | 5282160 | – | – | 72.17 |
| | Oseltamivir carboxylic acid | 100939439 | – | 61.55 | – |
| | mGTP | – | – | – | 106.44 |

*According to the literature [21,42] and

**[43].

**3.6.1. Phenolic compounds against active site of NA structure.** The five phenolic compounds, including gallic acid, methyl gallate, ethyl gallate, chlorogenic acid, afzelin, and myricitrin, can interact with the key amino acid residues present in the catalytic pocket of NA, which is crucial for viral spread to adjacent cells [45–47]. Among these, myricitrin (myricetin 3-rhamnoside) exhibited the greatest binding energy, forming twelve hydrogen bonds with eight amino acids within both the catalytic inner and outer shells of the NA structure, including Lys150 (2.24 Å), Asp151 (2.37 Å), Arg156 (2.69 Å), Trp179 (2.02 Å), Ser180 (2.56 Å and 3.00 Å), Glu278 (2.22 Å), Glu228 (2.42 Å and 2.60 Å), Arg293 (2.59 Å and 2.75 Å), and Arg368 (2.08 Å) (Fig 9). Its myricetin core also established hydrophobic interactions with Arg118, Lys150, Arg368, and Tyr402, ranging from 3.68 to 5.54 Å. This evidence aligns with Mothlhatlego et al. [48], who demonstrated that myricitrin, purified from *Newtonia buchananii*, inhibits various stages of the influenza A life cycle, including attachment and entry. Of similar significance, afzelin is anticipated to be functionally active due to its formation of nine hydrogen bonds with Glu119, Ile149, Arg152, Glu228, Arg225, Glu278, and Arg293 (1.79–2.68 Å). It is further stabilized in the target region by forming various hydrophobic interactions (π-cation, π-alkyl, and alkyl bonds) with Lys150, Arg152, Arg293, and Arg368, ranging from 3.12 to 4.87 Å. Apart from its higher energy level compared to the reference ligands (e.g., oseltamivir acid, quercetin, and isorhamnetin), our findings are further supported by recent study showing that *Thuja orientalis* folium extract, rich in myricetin, quercetin, quercitrin, amentoflavone, and afzelin, substantially inhibits influenza A virus (A/PR/8/34) attachment and entry by blocking neuraminidase activity [49]. Chlorogenic acid, with a fitness score of 58.45, can form ten promising hydrogen bonds with seven active residues, including Glu119 (1.80–2.37 Å), Gly147 (2.35 Å), Thr148 (3.01 Å), Arg152 (2.05 Å), Arg156 (2.96 Å), Trp179 (2.61 Å), Glu278 (1.81–2.77 Å), and Tyr402 (3.02 Å). Based on its caffeic-derived moiety, this phenolic can also create π-cation and π-lone pair interactions with Arg156 and Thr148, ranging from 2.69 to 3.44 Å. Ding et al. (2017) provided clear evidence to strengthen this finding, showing that chlorogenic acid acts as a neuraminidase blocker, effectively inhibiting influenza A virus in both in vitro and *in vivo* models. Recent study also indicated that the higher levels of chlorogenic acid in decaffeinated coffee beans, combined with lower caffeine levels, significantly impact their ability to inhibit neuraminidase [50]. Despite gallic acid and its two alkylated derivatives, methyl gallate and ethyl gallate, showing lower binding energies compared to oseltamivir carboxylic acid, they should be considered pharmacologically active due to their ability to interact with the catalytic residues of this viral surface protein, similar to the mechanism observed in anti-neuraminidase agents. Upon closer examination, gallic acid well established hydrogen bonds with Arg118 (1.94–2.54 Å), Glu119 (1.95 Å), and Arg368 (2.16 Å). Furthermore, the π-system within the benzene ring facilitated hydrophobic interactions with Arg293 (3.82 Å), Lys150 (3.82 Å), Glu278 (4.89 Å), and Tyr402 (2.35 Å) around the central ligand. A similar trend was observed with both methyl and ethyl gallate, which, in addition to forming hydrogen bonds, also created carbon-hydrogen bonds with Glu227 (2.90–3.05 Å). This led to higher binding energies (50.99 and 54.18) compared to gallic acid, their parent structure. For instance, ethyl gallate, the major constituent of in the *S. terebinthifolia* aqueous-ethanolic extract, well established six hydrogen bonds with Arg118 (1.93–2.54 Å), Glu119 (2.03 Å), Glu277 (3.05), and Arg368 (2.09 Å, interacting twice). Apart from the clear evidence provided by Zhang et al. [24], and Kaihatsu et al. [51] clearly demonstrated that ethyl gallate, isolated from coffee beans, exhibited anti-influenza activity with an $IC_{50}$ value greater than 2 μM, thus confirming our findings.

**3.6.2. Non-polyphenols against NA.** Previous research has demonstrated that both phenolic and non-phenolic substances can inhibit the viral replication cycle synergistically, although they may possess limited antiviral properties individually [52]. This underscores the significance of investigating compound interactions for potential therapeutic applications. In addition to the polyphenols, our study identified carboxylic acids such as suberic, azelaic, undecanedioic, dodecanedioic, and hexadecanedioic acids in the aqueous ethanolic extract, with fitness scores ranging from 46.74 to 61.95, exceeding those of gallic acid, quercetin, and rhamnetin (positive ligands). Although direct evidence concerning the anti-influenza activities of these carboxylic acids remains unclear, they show no violations of Lipinski's rule (as per the original guidelines specified by Pfizer), suggesting their potential drug-likeness properties (Table 2). Undecanedioic acid, for instance, is capable of forming five hydrogen bonds with the catalytic residues, including Lys150 (2.68 Å), Arg156

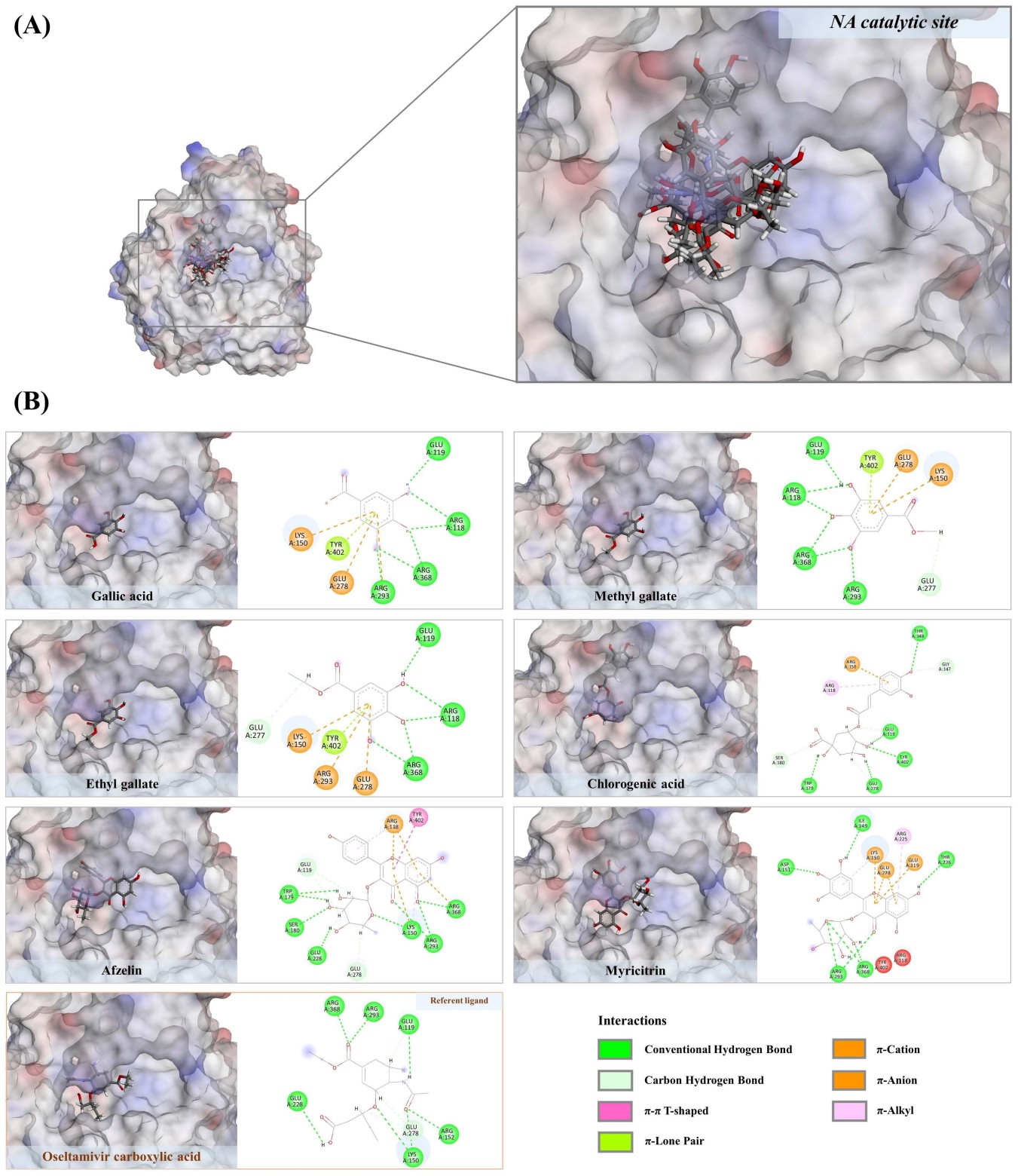

**Fig 9. A)** The three-dimensional structure depicting the structural alignment of polyphenols found in the *S. terebinthifolius* hydroethanolic extract (gallic acid, methyl gallate, ethyl gallate, chlorogenic acid, afzelin, and myricitrin) with the reference ligand drug (oseltamivir carboxylic acid) the active site of NA structure. **B)** The potential inhibitory effects of various phenolics towards the catalytic amino acid residues of NA structure.

(1.71 Å), Arg293 (6.29 Å), Ser439 (2.36 Å), and Arg368 (6.70 Å), using its carboxyl moieties. Additionally, it engages in three alkyl interactions with Arg118 (3.66–4.91 Å) and Lys150 (5.15 Å) through the alkyl moiety (Fig 10). Consequently, other carboxy acids are possible functionally active due to their similar binding patterns with the NA active region, where molecular dynamics simulation is primarily needed to address stability issues. Germacrone, with a binding energy of

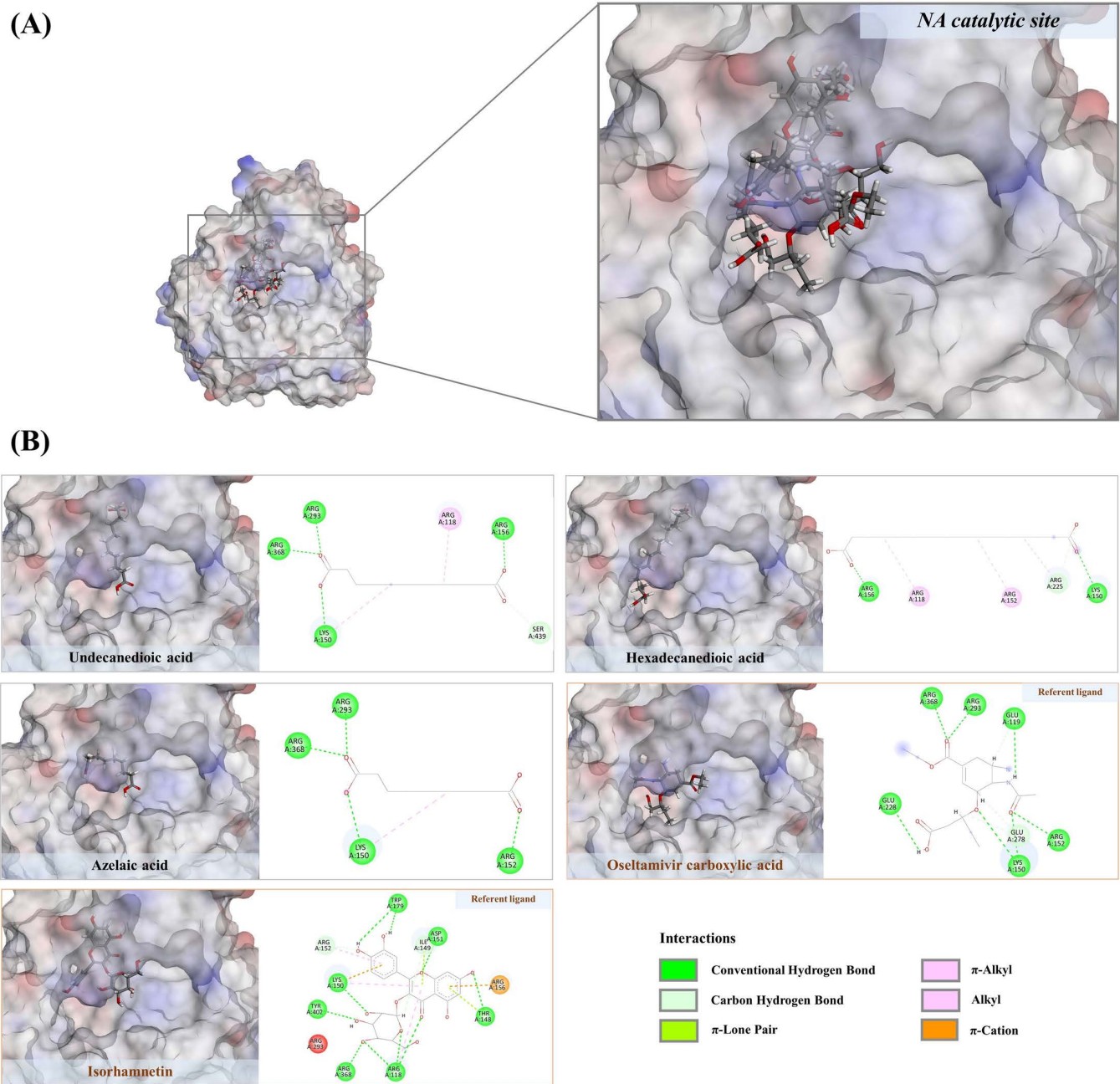

**Fig 10. A) Structural alignment of selected dicarboxylic acids (undecanedioic acid, hexadecanedioic acid and azelaic acid) exhibiting comparable binding energies with two reference ligands, oseltamivir carboxylic acid and Isorhamnetin. B)** Their 2D structures hypothesize potential inhibitory effects against the catalytic residues of the NA structure.

43.54, interacted with the NA catalytic region by forming a strong hydrogen bond with Arg152 (2.64 Å) and five prominent hydrophobic interactions with Arg152 (4.99 Å), Trp179 (4.74 Å), Ile223 (4.87 Å), and Arg225 (4.11–4.15 Å) (Fig 11). While MD simulation is still required, our findings are consistent with those of Liao et al. [29], which showed that germacrone from *Rhizoma curcuma* essential oil inhibits various stages of the influenza life cycle, including early stages and replication. This implies that neuraminidase is one of the target actions of the cyclic sesquiterpene observed herein.

### 3.7. Molecular Docking against cap-2 binding domain of influenza RNA polymerase

By mimicking the binding patterns of mGTP and favipiravir-RTP, certain simple phenolics and flavonoids have been well-documented to strongly interact with the highly conserved hydrophobic amino acid residues (e.g., Phe323, His357, Phe363, and Phe404) of the cap-2 binding domain of influenza A virus RNA polymerase (PB2), ultimately disrupting the replication processes of the flu viruses [43,53]. Apart from their moderate hydrophobicity (LogP between 0.21 and 1.84), which indicates their potential to penetrate the viral envelope (Table 2), virtual screening shows that the annotated metabolites can interact with the cap-2 binding domain, showing binding scores similar to those observed with the NA structure (Table 4). You et al. [22] demonstrated that gallic acid, isolated from the aqueous leaf extract of *Toona sinensis*, has the potential to inhibit influenza virus mRNA replication and MDCK plaque formation, in addition to functioning as a

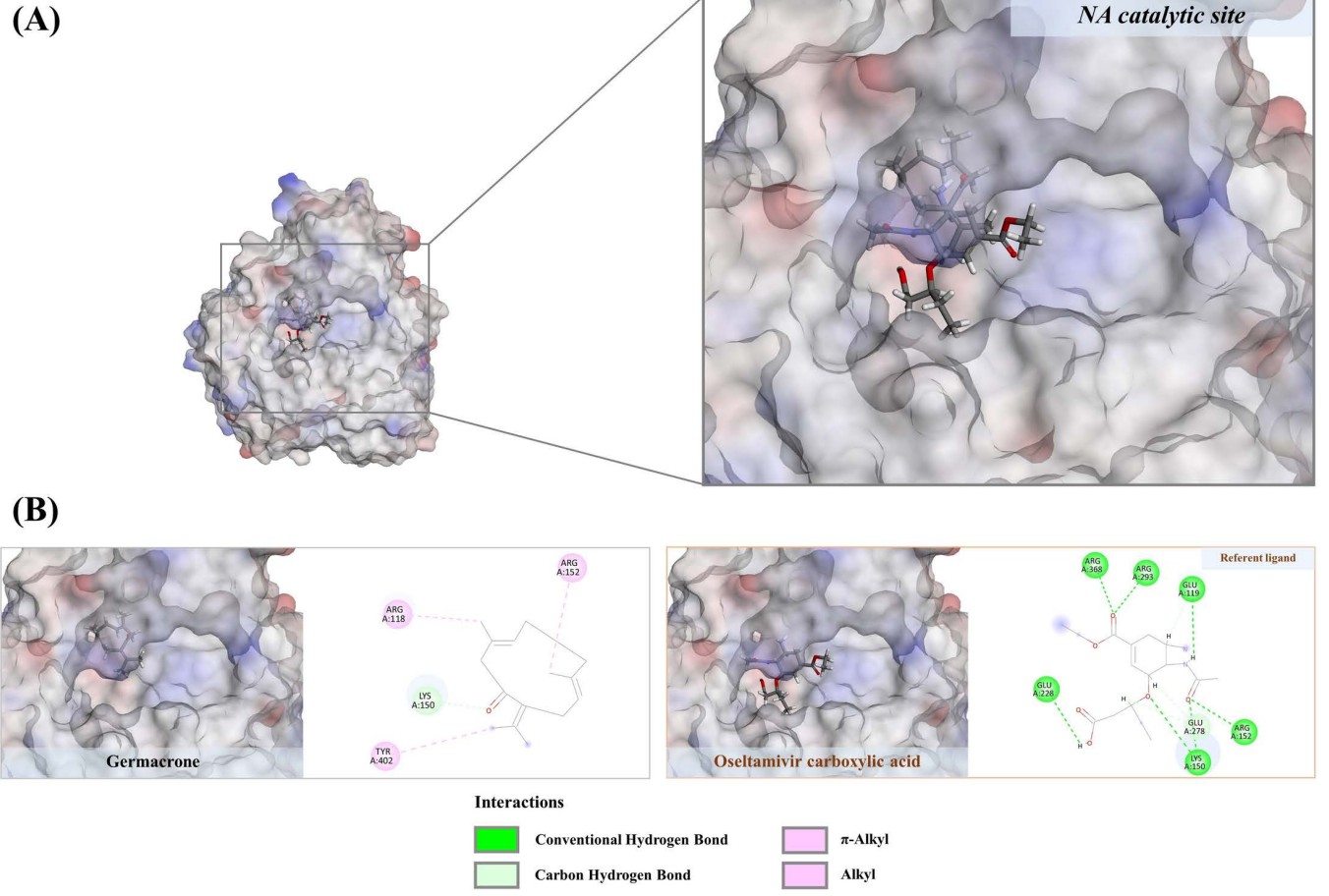

**Fig 11. A) Structural orientation of terpenoid derived natural products in drug-recognition site of neuraminidase.** Oseltamivir carboxylic acid and germacrone. **B)** Their 2D structures theorize potential inhibitory effects against the catalytic residues of the NA structure.

neuraminidase inhibitor. Therefore, the docked metabolites with fitness scores close to or higher than 42.71 are presumably active and are expected to strongly interact with the crucial hydrophobic residues of the cap-2 binding domain via extensive hydrophobic and hydrogen bond interactions, with bond distances not exceeding 3.00 Å and 5.00 Å, respectively. According to this criterion, 13 out of 18 docked metabolites were identified as having potential anti-influenza activities. Their molecular interactions are described in detail below.

### 3.7.1.  Phenolic compounds against PB2 subunit.

Six phenolic substances—gallic acid, methyl gallate, ethyl gallate, chlorogenic acid, afzelin, and myricitrin—demonstrated the ability to interact with crucial amino acid residues within the cap-binding domain, akin to those observed in positive ligands (e.g., mGTP, favipiravir-RTP, and quercetin) (Fig 12). Notably, they can engage π-stacking interactions with Phe363 and Phe404, which are essential for the cap-binding of influenza RNA polymerases. Additionally, they establish other interactions with neighboring aromatic amino acids, contributing to complex stabilization. Gallic acid, the third most abundant compound in the *S. terebinthifolia* extract and functioning as the benchmark ligand, is capable of creating three π-stacking interactions with Phe323 (5.30 Å), His357 (4.02 Å), and Phe404 (3.70 Å), while forming three rigid hydrogen bonds with Glu361 (2.10 Å), Lys376 (2.16 Å), and Phe404 (1.95 Å). Owing to further alkylation, methyl and ethyl gallate, with predicted logP values of 0.97 and 1.21, respectively, exhibited stronger binding properties to the cap-2 binding site of PB2 compared to their parent compound, gallic acid (Table 4). By means of its aromatic ring and methoxy group, the former reveals six promising hydrophobic interactions with crucial hydrophobic residues, including Phe323 (2.93–4.35 Å), His357 (3.43 Å), Phe363 (5.50 Å), Lys376 (5.44 Å), and Phe404 (3.67 Å), along with two hydrogen bonds with Lys375 (2.41–2.84 Å) in the target region. In the mGTP recognition site, the latter formed six hydrophobic interactions with His357 (3.49–5.04 Å), Phe323 (2.66–4.34 Å), Phe363 (5.04 Å), and Phe404 (3.59 Å), but more efficiently generated multiple hydrogen bonds with Ser337, Glu361, Lys376, Gln406, and Phe404, ranging from 1.83 to 2.86 Å. These results can be partially explained by previous findings, which demonstrated that plant extracts rich in gallic acid, and its two alkyl derivatives exert anti-influenza activities by disrupting the viral replication cycle, one of the key aspects of their molecular mode of action [51]. While chlorogenic acid is primarily known as a neuraminidase inhibitor [54], its polypharmacological effects on the cap-binding site should be comprehensively explored, given its promising binding energy and moderate hydrophobicity, superior to those of gallic acid and nearly comparable to quercetin (a positive ligand). Based on the caffeic acid-derived moiety, this metabolite generated three π-π stacked interactions with His357 (3.78 Å), Phe404 (3.80 Å), and Phe323 (5.74 Å), and formed multiple robust hydrogen bonds with Arg332 (2.17–5.89 Å), Ser337 (2.14–2.93 Å), Lys339 (2.80 Å), His357 (2.48 Å), and Phe404 (1.59 Å) in the mGTP and favipiravir-RTP recognition region. Though further experiments are essential, virtual screening conducted by Luo et al. [55] revealed that chlorogenic acid, mainly detected in *Lonicera japonica*, exerts a simultaneous effect against both NA and PB2, key proteins in the influenza life cycle, shedding light on the diverse effects of chlorogenic acid discussed here. Among the phenolics found in the *S. terebinthifolia* extract, myricitrin and afzelin displayed the highest binding energies, surpassing those of quercetin and hyperoside acting as positive ligands (Table 4). The initial one established numerous π-interactions with His357 (3.85–4.05 Å), Phe323 (2.75–3.53 Å), Phe363 (5.47 Å), and Phe404 (4.25 Å). In addition to three alkyl interactions with His357, Lys339, and Arg355 (3.98–4.88 Å) generated by the methyl group within its sugar moiety, five promising hydrogen bond interactions (1.74–2.97 Å) were also formed with neighboring amino acids, including Ser321, Ser324, Phe325, Arg355, and Glu361. A similar binding pattern was observed in afzelin, where it utilized kaempferol (aglycone) and its methyl group, creating various hydrophobic interactions with the highly conserved residues Lys339, Arg366, His357, Phe323, Phe363, and Phe404. Five rigid hydrogen bonds (1.70–2.89 Å) were also existing. The selective inhibition mechanism of myricitrin and afzelin towards influenza RNA polymerase remains ambiguous. However, a recent study demonstrated that myricetin, the aglycone core structure of myricitrin, with a fitness score of 60.99, can significantly inhibit influenza A virus replication. This is achieved by reducing influenza RNA polymerase activity through selective inhibition of the cap-2 binding domain (PB2) [42]. Owing to their structural similarity (characterized by a (C6-C3-C6) skeleton), we proposed that flavonoids derived from the phenylpropanoid pathway, such

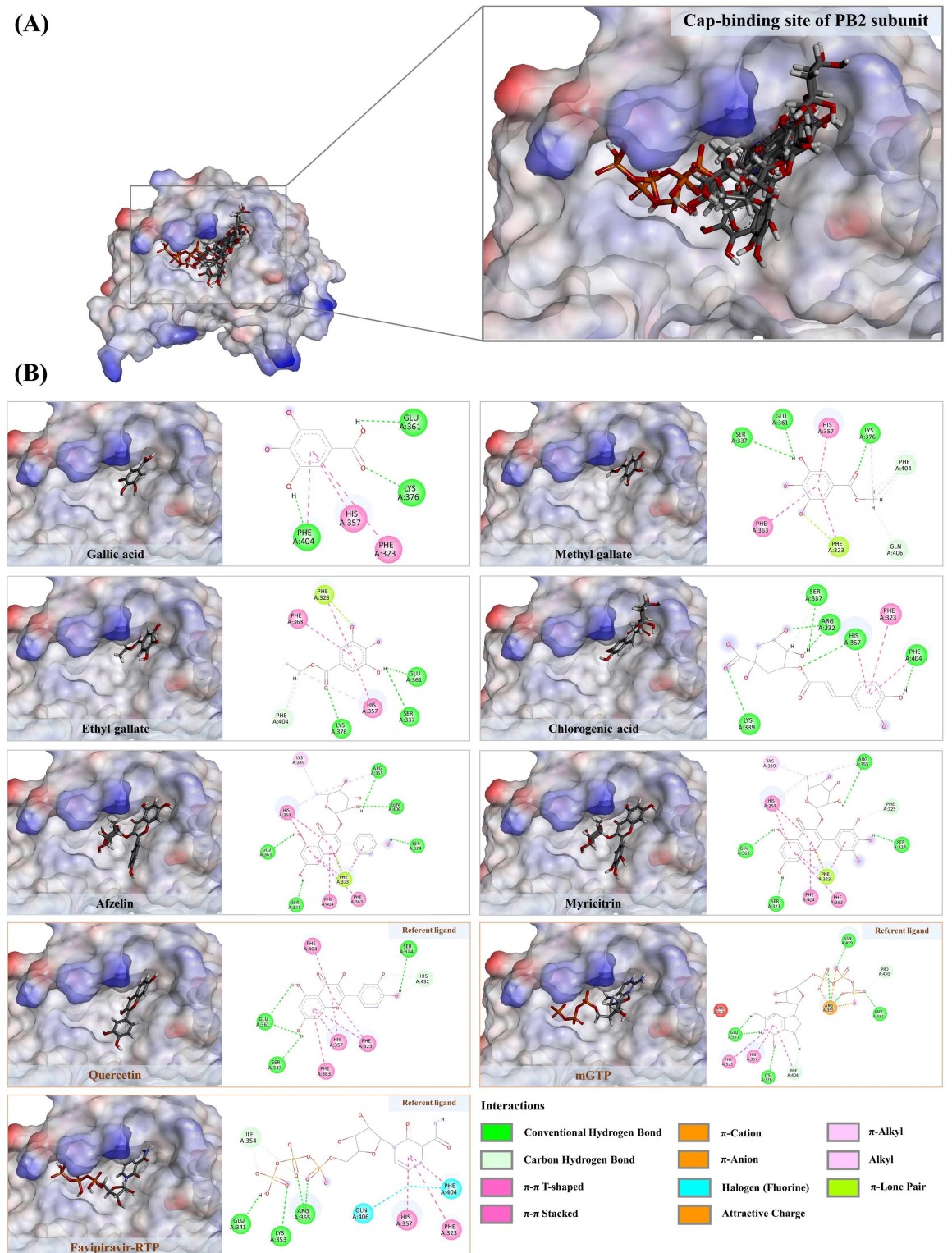

**Fig 12. A) Structural orientation of phenolic substances detected in the *S. terebinthifolia* hydroethanolic extract (gallic acid, methyl gallate, ethyl gallate, chlorogenic acid, afzelin, myricitrin) and three positive ligands (quercetin, mGTP, and favipiravir-RTP) in the cap-binding domain of PB2 subunit of influenza RNA polymerase. B)** 2D ligand-receptor interactions of chosen phenolic compounds with the PB2 subunit.

as myricitrin and afzelin, were likely active because they formed a greater number of hydrogen bonds and hydrophobic interactions with the mGTP/favipiravir-RTP recognition site.

### 3.7.2. Non-polyphenols against PB2 subunit.

Liao's research in 2013 clearly demonstrated that germacrone, a significant component of the essential oil in Rhizoma curcuma, not only influences the early stages of the influenza life cycle but also hinders its replication processes. Nonetheless, the mechanisms underlying its polypharmacological effects remain unclear. We observed that putative germacrone, tentatively identified with $m/z$ 219.1393 [M + H]⁺, can interact with the cap-2 binding site with an energy level comparable to that of the reference ligand, gallic acid (Fig 13A). Due to its strong hydrophobic properties (log P value = 2.93), germacrone formed eleven hydrophobic interactions—both alkyl-alkyl and π-alkyl—with highly conserved residues, including Phe323 (4.45 Å), Phe325 (4.47 Å), Arg355 (5.31 Å), His357 (3.56–4.79 Å), Phe363 (5.32 Å), Phe404 (3.89–4.73 Å), and Met431 (3.53–5.12 Å), suggesting a strong potential for binding with the mGTP binding module (Fig 13B). These findings prompted us to investigate the binding potential of dicarboxylic acids—suberic, azelaic, undecanedioic, dodecanedioic, and hexadecanedioic acids—owing to their promising lipophilicity (log P 1.15–2.98) and comparable binding energies with certain positive ligands, including gallic acid, quercetin, myricetin, and hyperoside (Table 4). Hexadecanedioic acid, for instance, established eleven promising hydrophobic interactions with crucial residues of the PB2 subunit, including Phe323 (4.67 Å), Arg355 (3.96–5.07 Å), His357 (3.44–4.72 Å), Met431 (4.96, 4.93, 5.38 Å), Phe363 (5.49 Å), and Phe404 (4.15–4.58 Å), via its alkyl moiety. Its two carboxyl groups also played a crucial role in forming five hydrogen bonds with Ser321 (1.76 Å), Arg332 (2.54 Å), Glu341 (2.26 Å), Lys353 (3.25 Å), and Ile354 (2.43 Å) (Fig 13B). A similar binding pattern was observed in other dicarboxylic acids. Due to a lack of evidence regarding the anti-influenza effects of various chain-length dicarboxylic acids, it is essential to employ alternative strategies, such as molecular dynamics simulations, to assess stability in different query complex structures.

## 3.8. Molecular dynamics simulation analysis

A 100 ns molecular dynamics (MD) simulation was performed to evaluate the stability of interactions between the target proteins PB2 and NA and their top three ligands chosen from the *S. terebinthifolia* aqueous ethanolic extract, validating the docking results (Figs 14 and 15). Regarding the PB2–ligand complexes, myricitrin (1st), afzelin (2nd), hexadecanedioic acid (3rd), and quercetin (control ligand) were found to properly fit within the cap-binding site of the PB2 subunit, with respective RMSD values of 1.30, 1.92, 1.68, and 1.55 Å, indicating sufficient stability of each protein-ligand complex throughout the simulation (Fig 14A). The average RMSD values for the NA protein with the selected ligands—afzelin (2nd), hexadecanedioic acid (3rd), undecanedioic acid (4th), and quercetin (control ligand)—were 1.71, 1.63, 1.55, and 1.79 Å, respectively (Fig 15A). Despite slight fluctuations observed during the simulation, the results indicated overall stability, with the complexes maintaining consistently low RMSD values under the given conditions. Protein residue flexibility was also analyzed through root mean square fluctuation (RMSF), which measures the positional changes of each residue as it interacts with its ligand over the simulation period. Obviously, the four ligands—myricitrin, afzelin, hexadecanedioic acid, and quercetin—showed promising RMSF values of 0.12, 0.14, 0.13, and 0.13 nm, respectively, while remaining bound to the cap-binding residues (Phe323, His357, Phe363, and Phe404), with low fluctuations observed throughout the 100 ns simulation (Fig 14B). Similarly, the average RMSF values for the NA protein with afzelin, hexadecanedioic acid, undecanedioic acid, and quercetin (control) were 0.11, 0.10, 0.11, and 0.10 nm, respectively, suggesting only slight fluctuations in the individual amino acid residues of this viral surface protein (Fig 15B). These results indicate that all protein–ligand complexes involving the two viral target proteins are structurally stable, with minimal fluctuations in the core binding regions of conserved amino acids, while remaining consistently positioned within the drug recognition sites. Equally important, both NA and PB2 exhibited lower radius of gyration and solvent-accessible surface area (SASA) values (Figs 14D and 15D), clearly indicating that the protein–ligand complexes maintained a stable conformation with no significant alterations under solvent-exposed conditions. PB2–ligand complexes displayed Rg values of 1.60, 1.61, 1.60, and 1.60 nm for myricitrin, afzelin, hexadecanedioic acid, and quercetin in that order; NA–ligand complexes exhibited values of 1.99, 1.97, 1.98, and

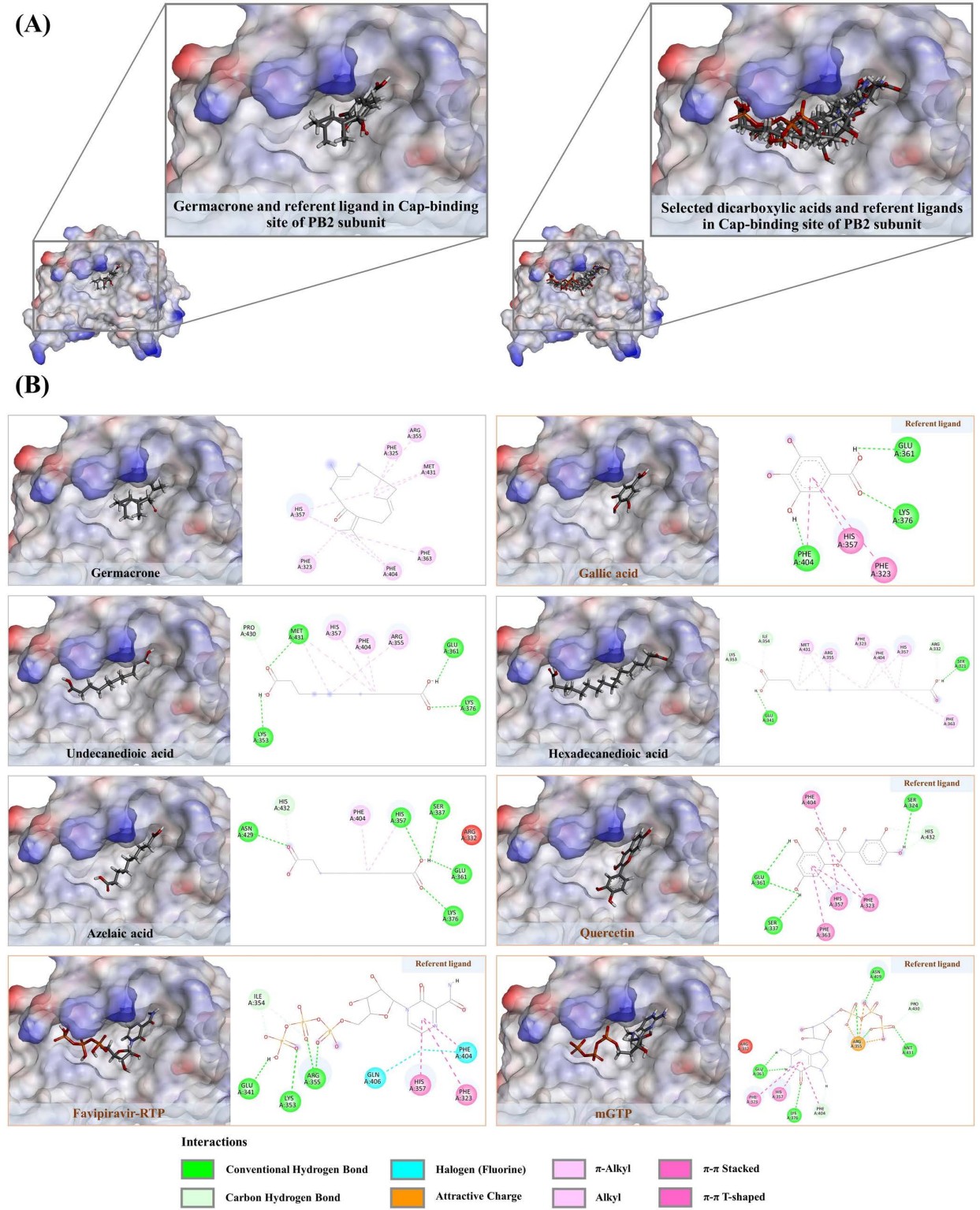

**Fig 13. A) Structural orientation of non-phenolic substances detected in the *S. terebinthifolia* aqueous ethanol extract with positive ligands lining in the cap-binding domain of PB2 subunit of influenza RNA polymerase.** Germacrone, referent ligand – Gallic acid, hexadecanedioic acid, undecanedioic acid, azelaic acid, referent ligands – quercetin, favipiravir-RTP, and mGTP. **B)** 2D ligand-receptor interactions of selected metabolites with the PB2 subunit.

**(A)**

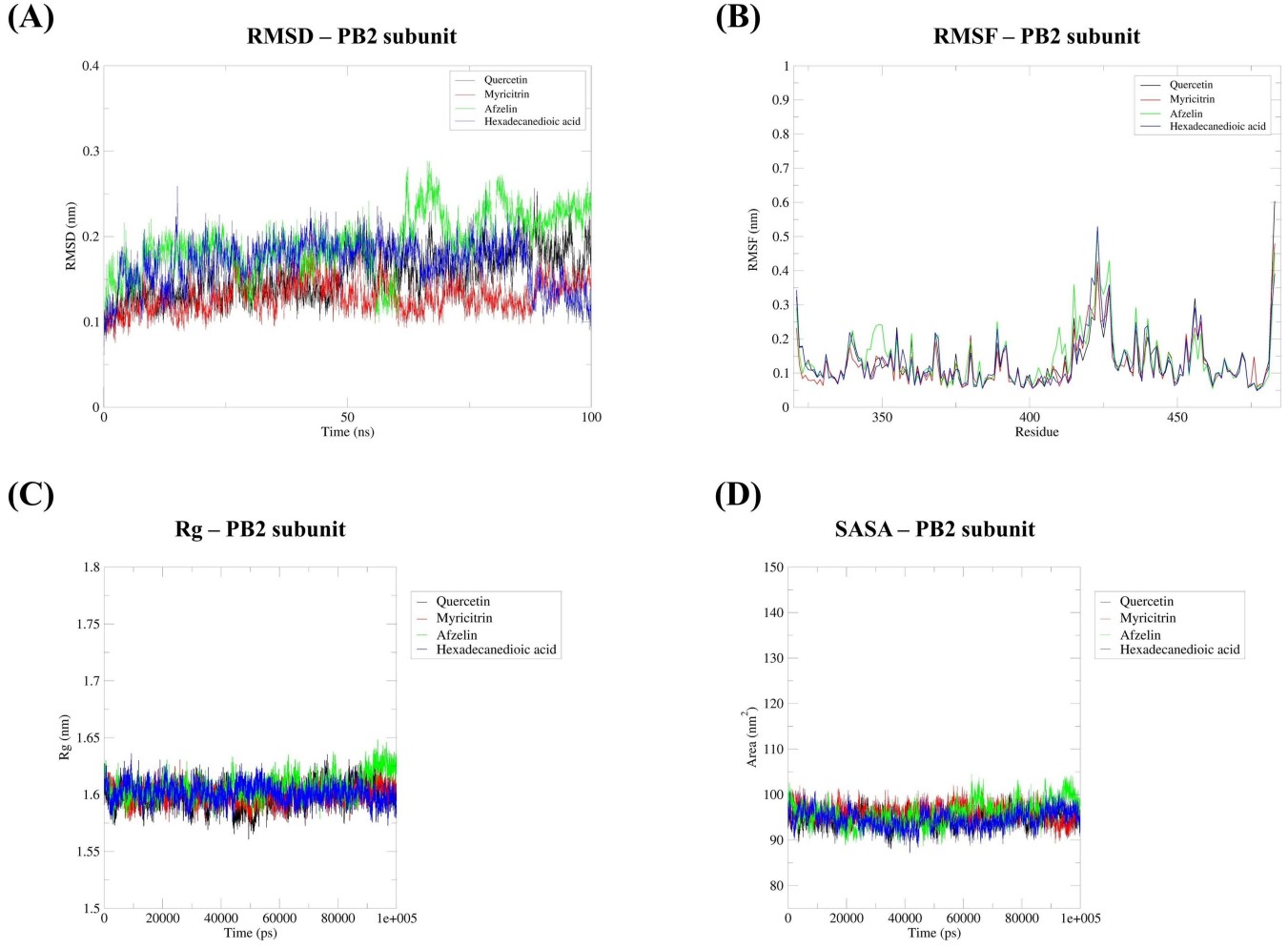

**Fig 14. MD simulation analysis of PB2 subunit with selected metabolites from hydroethanolic *S. terebinthifolia* extract (quercetin (control), myricitrin, afzelin, hexadecanedioic acid). A)** RMSD. **B)** RMSF. **C)** Radius of Gyration. **D)** Solvent accessible surface area.

1.97 nm for afzelin, hexadecanedioic acid, undecanedioic acid, and quercetin accordingly (Figs 14C and 15D). Overall, our selected phytochemicals from the hydroethanolic extract are structurally stable, as indicated by negligible fluctuations in various parameters measured at the druggable targets of the two influenza proteins.

## 4. Conclusion

As a new finding, our study indicates that the hydroethanolic extract (30% v/v ethanol solution) prepared from the aerial parts of *S. terebinthifolia* is a promising source of antioxidant and anti-influenza agents. Additionally, the newly established plant extract was found to be free of hemolytic activity, suggesting that it may be a safe antiviral agent. Gallic acid could be representatively detected from this phytochemical extract, according to RP-HPLC analysis. Notably, the use of high-resolution LC-ESI(±)-QTOF-MS/MS, with the aid of cheminformatics software, revealed that it contains not only gallic acid but also other relevant metabolites in the shikimate and phenylpropanoid pathways, such as methyl gallate, ethyl gallate, quinic acid, shikimic acid, chlorogenic acid, myricitrin, and afzelin. Additionally, dicarboxylic acids and a putative germacrone were also identified. The selected phytochemicals from the

**(A)**

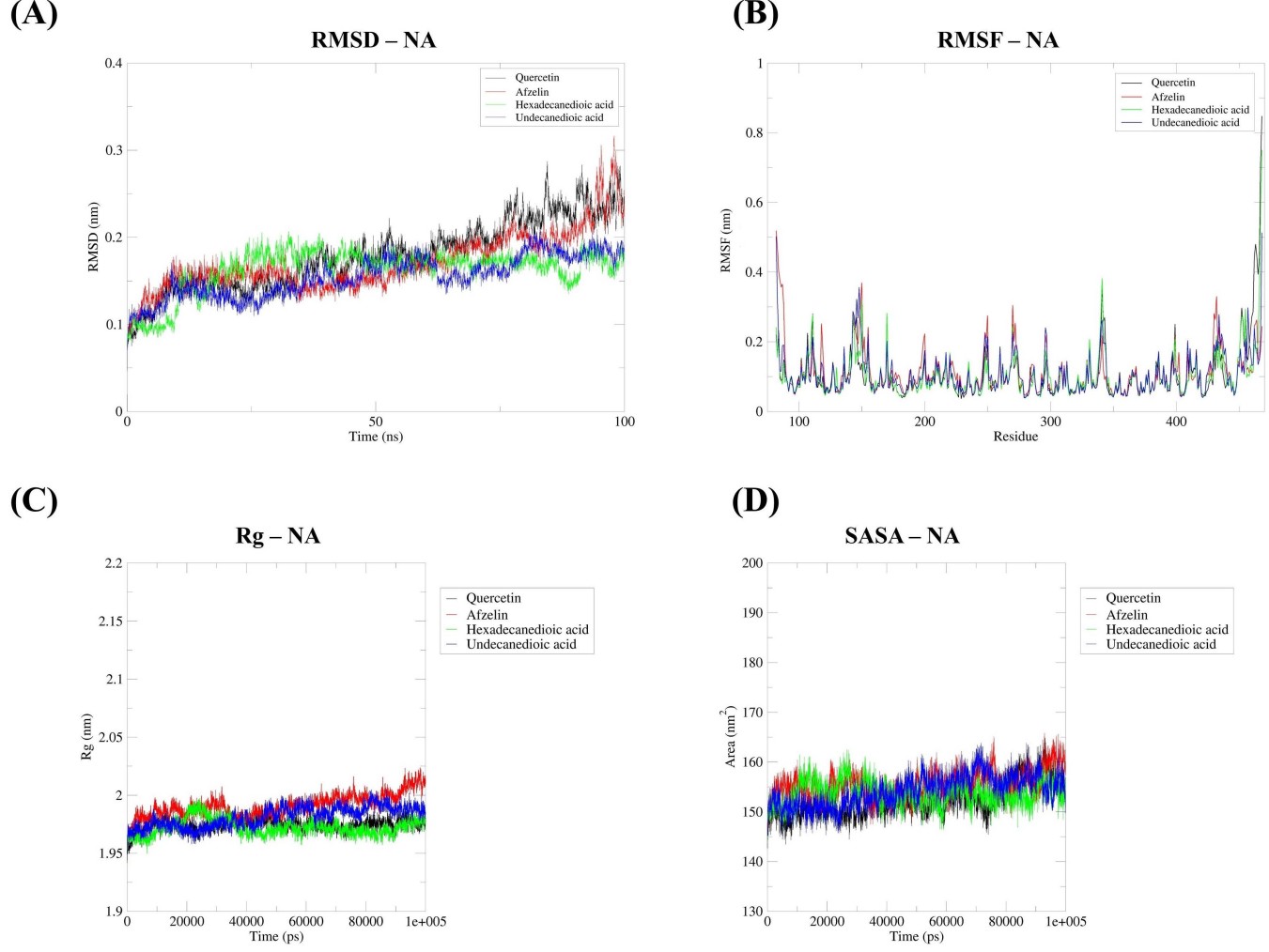

Fig 15. **MD simulation analysis of NA protein with selected metabolites from hydroethanolic** *S. terebinthifolia* **extract (quercetin (control), afzelin, hexadecanedioic acid, undecanedioic acid). A)** RMSD. **B)** RMSF. **C)** Radius of Gyration. **D)** Solvent accessible surface area.

hydroethanolic extract demonstrated strong drug-likeness, aligning with Lipinski's and Ghose's rules, thereby indicating their therapeutic potential. *In silico* docking studies identified neuraminidase (NA) and the cap-binding domain of influenza RNA polymerase (PB2) as promising inhibitory targets, supporting the extract's antiviral activity. Molecular docking and MD simulations further confirmed the binding stability of representative natural product classes within the extract. Given the broad-spectrum antiviral properties of phenolic compounds, particularly against various influenza pathways, ongoing investigations are assessing the extract's efficacy against H5N1—a strain currently posing a global health threat.

## Acknowledgments

The authors are extremely grateful to the Faculty of Pharmaceutical Sciences, Khon Kaen University, Thailand. We also would like to thank Kiattawee Choowongkomon from the Department of Biochemistry, Faculty of Sciences, Kasetsart University, Bangkok, Thailand, for providing the GOLD program used in molecular docking analyses.

## Author contributions

**Conceptualization:** Napapuch Nopkuesuk, Anuwatchakij Klamrak, Poramet Sitthiwong, Nisachon Jangpromma, Sirinan Kulchat, Kiattawee Choowongkomon, Rina Patramanon, Arunrat Chaveerach, Samaporn Teeravechyan, Jureerut Daduang, Sakda Daduang.

**Data curation:** Napapuch Nopkuesuk, Anuwatchakij Klamrak, Jaran Nabnueangsap, Jaraspim Narkpuk, Yutthakan Saengkun, Piyapon Janpan, Kiattawee Choowongkomon.

**Formal analysis:** Napapuch Nopkuesuk, Anuwatchakij Klamrak.

**Funding acquisition:** Sakda Daduang.

**Investigation:** Napapuch Nopkuesuk, Anuwatchakij Klamrak, Jaran Nabnueangsap, Jaraspim Narkpuk, Shaikh Shahinur Rahman, Yutthakan Saengkun, Piyapon Janpan, Thananya Soonkum, Jureerut Daduang.

**Methodology:** Napapuch Nopkuesuk, Anuwatchakij Klamrak, Jaran Nabnueangsap, Jaraspim Narkpuk, Yutthakan Saengkun, Piyapon Janpan, Thananya Soonkum.

**Project administration:** Sakda Daduang.

**Resources:** Samaporn Teeravechyan.

**Supervision:** Sakda Daduang.

**Validation:** Anuwatchakij Klamrak, Sakda Daduang.

**Visualization:** Napapuch Nopkuesuk, Anuwatchakij Klamrak, Shaikh Shahinur Rahman, Yutthakan Saengkun, Piyapon Janpan.

**Writing – original draft:** Napapuch Nopkuesuk, Anuwatchakij Klamrak, Shaikh Shahinur Rahman.

**Writing – review & editing:** Napapuch Nopkuesuk, Anuwatchakij Klamrak, Shaikh Shahinur Rahman.

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
