## [Decision Letter · Decision Letter 0]

6 Feb 2025

PONE-D-24-35830Hydroethanolic extract of Schinus terebinthifolia as a promising source of anti-influenza agents: Phytochemical profiling, cheminformatics, and In silico docking assessmentsPLOS ONE

Dear Dr. Daduang,

Thank you for submitting your manuscript to PLOS ONE. After careful consideration, we feel that it has merit but does not fully meet PLOS ONE’s publication criteria as it currently stands. Therefore, we invite you to submit a revised version of the manuscript that addresses the points raised during the review process.

We look forward to receiving your revised manuscript.

Kind regards,

A.H.M. Khurshid Alam, Ph.D

Academic Editor

PLOS ONE

Comments from PLOS Editorial Office: We note that one or more reviewers has recommended that you cite specific previously published works. As always, we recommend that you please review and evaluate the requested works to determine whether they are relevant and should be cited. It is not a requirement to cite these works. We appreciate your attention to this request.

Journal Requirements:

“This research was funded by the Program Management Unit for Human Resources and Institutional Development, Research and Innovation (PMU-B) for postmaster scholarship. It received partial funding from The Fundamental Fund of Khon Kean University (KKU), which received financial support from the National Science, Research and Innovation Fund (NSRF), Thailand. The research was supported by NSRF under the Basic Research Fund of Khon Kaen University.”

“NO authors have competing interests”

6. Please provide a complete Data Availability Statement in the submission form, ensuring you include all necessary access information or a reason for why you are unable to make your data freely accessible. If your research concerns only data provided within your submission, please write "All data are in the manuscript and/or supporting information files" as your Data Availability Statement.

7. When completing the data availability statement of the submission form, you indicated that you will make your data available on acceptance. We strongly recommend all authors decide on a data sharing plan before acceptance, as the process can be lengthy and hold up publication timelines. Please note that, though access restrictions are acceptable now, your entire data will need to be made freely accessible if your manuscript is accepted for publication. This policy applies to all data except where public deposition would breach compliance with the protocol approved by your research ethics board. If you are unable to adhere to our open data policy, please kindly revise your statement to explain your reasoning and we will seek the editor's input on an exemption. Please be assured that, once you have provided your new statement, the assessment of your exemption will not hold up the peer review process.

8. PLOS requires an ORCID iD for the corresponding author in Editorial Manager on papers submitted after December 6th, 2016. Please ensure that you have an ORCID iD and that it is validated in Editorial Manager. To do this, go to ‘Update my Information’ (in the upper left-hand corner of the main menu), and click on the Fetch/Validate link next to the ORCID field. This will take you to the ORCID site and allow you to create a new iD or authenticate a pre-existing iD in Editorial Manager.

9. Please amend the manuscript submission data (via Edit Submission) to include authors Napapuch Nopkuesuk, Anuwatchakij Klamrak, Jaran Nabnueangsap, Jaraspim Narkpuk, Shaikh Shahinur Rahman, Yutthakan Saengkun, Piyapon Janpan, Thananya Soonkum, Poramet Sitthiwong6, Nisachon Jangpromma, Sirinan Kulchat, Kiattawee Choowongkomon, Rina Patramanon, Arunrat Chaveerach, Samaporn Teeravechyan and Jureerut Daduang.

10. Please include your full ethics statement in the ‘Methods’ section of your manuscript file. In your statement, please include the full name of the IRB or ethics committee who approved or waived your study, as well as whether or not you obtained informed written or verbal consent. If consent was waived for your study, please include this information in your statement as well.

Reviewers' comments:

Reviewer's Responses to Questions

**Comments to the Author**

1. Is the manuscript technically sound, and do the data support the conclusions?

Reviewer #1: Partly

Reviewer #2: Yes

2. Has the statistical analysis been performed appropriately and rigorously? 

Reviewer #1: Yes

Reviewer #2: Yes

3. Have the authors made all data underlying the findings in their manuscript fully available?

Reviewer #1: Yes

Reviewer #2: Yes

4. Is the manuscript presented in an intelligible fashion and written in standard English?

Reviewer #1: Yes

Reviewer #2: Yes

5. Review Comments to the Author

Reviewer #1: The manuscript titled "Hydroethanolic extract of Schinus terebinthifolia as a promising source of anti-influenza agents" presents interesting findings regarding the antiviral properties of the plant extract. However, several areas could be improved for clarity, depth, and overall impact. Here are some comments:

1. The introduction should provide a clearer rationale for the study. While it mentions the biological activities of Schinus terebinthifolia, it could benefit from a more detailed discussion on the significance of exploring its anti-influenza properties specifically. This would help contextualize the research better for readers unfamiliar with the subject.

2. The methods section lacks sufficient detail regarding the extraction process and the specific conditions used (e.g., temperature, duration). Providing this information would enhance reproducibility and allow other researchers to replicate the study more effectively.

3. The results could be presented more clearly. The HPLC analysis results for gallic acid should include a more comprehensive discussion on the implications of the yield (1.71 mg/g DW) and how it compares to other studies. Additionally, including visual aids such as graphs or tables would help in conveying the data more effectively.

4. The discussion section should delve deeper into the implications of the findings. While it mentions the potential of gallic acid and other metabolites as anti-influenza agents, it could explore the mechanisms of action in more detail. This would provide a better understanding of how these compounds interact with viral targets like neuraminidase and PB2.

5. The manuscript should explicitly state the limitations of the study, such as the in vitro nature of the findings and the need for in vivo studies to confirm efficacy.

6. Molecular dynamic studies shall be performed for top 03 ligands for 100ns to confirm the stability.

7. Authors are suggested to add following references;

https://doi.org/10.1016/j.heliyon.2023.e15952

https://doi.org/10.1080/14786419.2021.2013839

https://doi.org/10.1016/j.sajb.2023.01.005

https://doi.org/10.1080/10496475.2019.1689542

Reviewer #2: Please add more details in introduction part to show research impact.

provide justification for selecting Ethanol

improve images quality (Special HPLC results)

Please add the catalog number for standard reference for HPLC chemicals used in the recent study.

improve conclusion.

6. PLOS authors have the option to publish the peer review history of their article (what does this mean? ). If published, this will include your full peer review and any attached files.

**Do you want your identity to be public for this peer review?** For information about this choice, including consent withdrawal, please see our Privacy Policy .

Reviewer #1: No

Reviewer #2: No

---

## [Author Response · Author response to Decision Letter 1]

22 Apr 2025

Dear Editor and Reviewers,

Firstly, we would like to inform you that we have decided to revise the title from “Hydroethanolic extract of Schinus terebinthifolia as a promising source of anti-influenza agents: Phytochemical profiling, cheminformatics, and in silico docking assessments” to “Hydroethanolic extract of Schinus terebinthifolia as a promising source of anti-influenza agents: Phytochemical profiling, cheminformatics, and in silico molecular docking simulations”, to better reflect the inclusion of molecular dynamics simulation studies.

Secondly, thank you very much to editor and reviewers for the revision of our manuscript title “Hydroethanolic extract of Schinus terebinthifolia as a promising source of anti-influenza agents: Phytochemical profiling, cheminformatics, and in silico molecular docking simulations”. We would like to explain point-by-point the details of the revisions in the manuscript and our responses to the reviewers' comments.

The revisions that have been made to the manuscript were provided in the "Response to Reviewers - PONE-D-24-35830 file"

---

## [Editor Report · Decision Letter 1]

6 May 2025

Hydroethanolic Extract of Schinus terebinthifolia as a Promising Source of Anti-Influenza Agents: Phytochemical Profiling, Cheminformatics, Molecular Docking and Dynamics Simulations

PONE-D-24-35830R1

Dear Dr. Sakda,

We’re pleased to inform you that your manuscript has been judged scientifically suitable for publication and will be formally accepted for publication once it meets all outstanding technical requirements.

Kind regards,

A.H.M. Khurshid Alam, Ph.D

Academic Editor

PLOS ONE
---

## [Editor Report · Acceptance letter]

PONE-D-24-35830R1

PLOS ONE

Dear Dr. Daduang,

I'm pleased to inform you that your manuscript has been deemed suitable for publication in PLOS ONE. Congratulations! Your manuscript is now being handed over to our production team.

Kind regards,

on behalf of

Dr. A.H.M. Khurshid Alam

Academic Editor

PLOS ONE